# Efficient Masked AutoEncoder for Video Object Counting and a Large-Scale Benchmark

**Bing Cao, Quanhao Lu, Jiekang Feng, Qilong Wang, Qinghua Hu, Pengfei Zhu**[*]
Tianjin University
`{caobing,luquanhao,fengjiekang,qlwang,huqinghua,zhupengfei}`
`@tju.edu.cn`

## Abstract

The dynamic imbalance of the fore-background is a major challenge in video object counting, which is usually caused by the sparsity of target objects. This remains understudied in existing works and often leads to severe under-/over-prediction errors. To tackle this issue in video object counting, we propose a density-embedded Efficient Masked Autoencoder Counting (E-MAC) framework in this paper. To empower the model's representation ability on density regression, we develop a new `Density-Embedded Masked mOdeling (DEMO)` method, which first takes the density map as an auxiliary modality to perform multimodal self-representation learning for image and density map. Although `DEMO` contributes to effective cross-modal regression guidance, it also brings in redundant background information, making it difficult to focus on the foreground regions. To handle this dilemma, we propose an efficient spatial adaptive masking derived from density maps to boost efficiency. Meanwhile, we employ an optical flow-based temporal collaborative fusion strategy to effectively capture the dynamic variations across frames, aligning features to derive multi-frame density residuals. The counting accuracy of the current frame is boosted by harnessing the information from adjacent frames. In addition, considering that most existing datasets are limited to human-centric scenarios, we first propose a large video bird counting dataset, *DroneBird*, in natural scenarios for migratory bird protection. Extensive experiments on three crowd datasets and our *DroneBird* validate our superiority against the counterparts. The code and dataset are available [1].

## 1 Introduction

Video object counting aims to estimate the number of objects in video scenes and has been used in various practical applications, from traffic management to public security. It has the potential to be used in decreasing the workload of public management and protecting migratory birds. Due to the crucial role of object counting in multiple application scenarios, it has attracted broad attention in recent years with the development of computer vision.

Despite many excellent works that have been proposed over the past decades, most of these methods are based on static single-frame images (Zhang et al., 2016; Li et al., 2018) extracted from video, leading to significant loss of dynamic inter-frame information, especially for the swiftly moving targets. In practice, such as crowd or animal activity analysis, the source data is often captured in video form by surveillance cameras or drones. Unlike static single-frame images, video data is significantly dynamic in the spatial motion variations of foreground objects across adjacent time instances, thereby providing richer contextual information. Therefore, by capturing the inter-frame information between video frames, the model is qualified to better perceive dynamical targets, thereby improving the accuracy and stability of the counting performance. To this concern, some video counting methods appeared recently (Zou et al., 2019; Bai & Chan, 2021; Hossain et al., 2020), which aim to capture the dynamism between frames by employing techniques such as 3D convolutions or incorporating additional information. However, since the video data inherently suffers from the problem

---

[*]Corresponding author
[1]https://github.com/mast1ren/E-MAC

of redundant background information Zhou et al. (2022), extracting features of dynamic targets in multiple frames may lead to an imbalance between foreground and background information, posing challenges for the model's optimization and inference.

More recently, inspired by the unprecedented strong self-representation ability of pre-trained vision foundational models (He et al., 2021; Tong et al., 2022), researchers have injected these foundation models into downstream vision tasks to fully exploit their representational potential. Inspired by this, we present an **E**fficient **M**asked **A**utoencoder **C**ounting (E-MAC) framework for video object counting. Our E-MAC introduced optical flow-based Temporal Collaborative Fusion (TCF) to establish inter-frame relationships, constructing a pre-trained visual foundation model-based video counting framework. The optical flow between frames is used to warp the predicted density map of the adjacent frame to the current frame. Then, we perform cross-attention between the warped density map and the predicted current density map to get the final results.

However, the high dynamic of the video data often leads to imbalanced optimization of the sparse foreground. Different from most existing techniques, we take the density map as an additional auxiliary modality of images and transfer the self-representation foundation model to object counting for the first time. We constructed a Density-Embedded Masked mOdeling (DEMO) that takes inputs from both the image and the density map, which performs feature interaction through the encoder and reconstructs the density map from masked image and density map. To this end, the density self-representation learning drives the regression implicitly by reconstructing the masked density maps. In addition, while the self-representation learning of density maps facilitates efficient density regression, the dynamic nature of foreground objects in video data still brings significant imbalanced challenges to optimization. Stochastic masked image modeling struggles to focus the model on extracting features from dynamic moving targets, leading to redundant background reconstruction that hinders model optimization. To handle this dilemma, we further develop a Spatial Adaptive Masking (SAM) to generate dynamic efficient masks. During training, SAM dynamically generates masks based on the correlated density map of each sample, providing valid information while filtering out redundant background details. Our framework employs a post-fusion strategy and develops a simple cross-attention module to compute the residuals between adjacent predicted density maps, and design a skip connect to add the residuals to the predicted density map of the current frame, which ultimately filters the non-dynamic objects in the background.

In this paper, we validate our E-MAC not only in human-centric scenarios but also in natural scenarios. A large-scale video bird counting dataset *DroneBird* is collected for migratory bird protection. To the best of our knowledge, DroneBird is the first video bird counting dataset that is captured from a drone's viewpoint and provides abundant annotations and rich attributions. Experimental results on three human-centric datasets and our DroneBird dataset demonstrate the superiority of our method over the competing methods. Our main contributions are summarized as follows:

- We propose a density-embedded efficient masked autoencoder counting framework for video object counting, which integrates the foundational model and takes the density map as an auxiliary modality to perform self-representation learning, effectively driving density map regression implicitly.

- We propose an efficient spatial adaptive masking method to overcome the dynamic density distribution and make the model focus on the foreground regions. It adaptively generates image masks according to the corresponding density maps, effectively addressing the problem of imbalanced fore-background.

- We propose a large-scale bird counting dataset *DroneBird* for bird activities analysis. To our knowledge, DroneBird is the first video bird counting dataset. Extensive experiments on three human-centric scenarios and our DroneBird dataset validate our superiority compared to the competing methods.

## 2 RELATED WORK

**Object Counting.** The vast majority of proposed object counting methods were commonly based on a single image. Existing counting methods (Li et al., 2018; Liu et al., 2019; Liang et al., 2022) were mainly based on density map estimation, which generated the density map from point annotations and took it as the ground truth. Most current counting methods tend to use density map

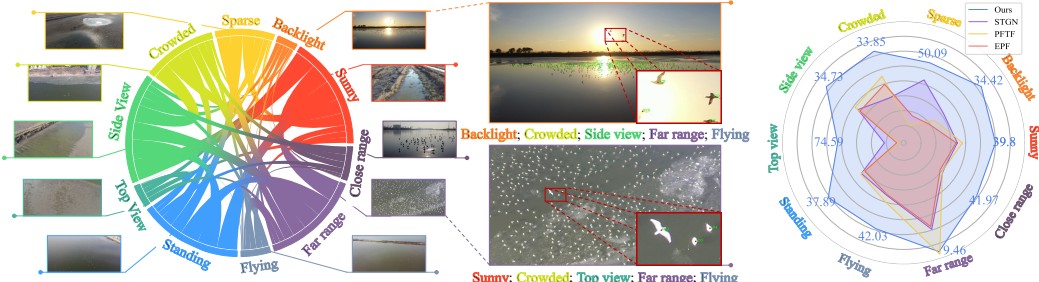

Figure 1: The chord diagram illustrates the associations between various attributes of our proposed dataset. Each attribute showcases a portion of the dataset's examples as references. We provide two zoomed-in examples for better visualization. The right part represents the experimental result of our proposed method and previous video counting method on each attribute of our DroneBird dataset.

regression as the pretext task of object counting since it provides more low-level supervision signals and is easier to optimize. Earlier researchers (Zhang et al., 2016; Li et al., 2018) explored improving convolutional neural network structures to enhance density regression performance by extracting multi-scale features from images. Recent methods (Ma et al., 2019; Lin et al., 2022) utilized Bayesian loss for density contribution models from point labeling, improving upon density map supervision. Additionally, researchers have integrated CNNs and Transformers to leverage the attention mechanism (Tian et al., 2021; Liang et al., 2022). More recently, some methods introduced pre-trained foundational models to build object counting methods (Jiang et al., 2023; Kang et al., 2024), thereby counting the number of any examples. This inspired us to explore the visual foundation model based video object counting framework.

**Video Object Counting.** The single-frame image methods focus on spatial information from static images and neglect temporal processing, making it difficult to address the dynamic nature of video object counting tasks. The target of video object counting is to predict the number of objects in each frame of the video. For the evaluation of counting results, video counting calculates the difference between the predicted results and the ground truth for each frame, and then computes the mean absolute error (MAE) and mean squared error (RMSE) across all frames. Video object counting methods aim to leverage information from neighboring frames to enhance the estimation of the current frame. LSTM and 3D convolutions are commonly used methods for modeling temporal dependencies between frames (Zou et al., 2019; Shi et al., 2015). Unlike these implicit methods of establishing frame associations, leveraging object movement direction and optical flow information can further enhance counting accuracy (Zhu et al., 2021; Hou et al., 2023). However, existing video counting methods (Liu et al., 2020; Hou et al., 2023) mainly address temporal relationships but often neglect intra-frame dynamics of foreground regions. Additionally, the high cost of dot annotations restricts the availability of large video counting datasets, complicating effective learning of dynamic regions. Our proposed method treats counting as a density reconstruction task, incorporating self-representation learning of density maps with a dynamic spatial adaptive masking module, which significantly enhances the counting performance.

**Masked Image Modeling.** Masked image modeling refers to the reconstruction of the masked portion of a masked image by learning its representation. With the application of Transformer (Vaswani et al., 2017) in vision and the success of the BERT (Devlin et al., 2019) pre-training paradigm in natural language processing in recent years, masked image modeling has achieved great progress. After some enlightening work (Vincent et al., 2008; Chen et al., 2020; Bao et al., 2022), MAE (He et al., 2021) chunks the image, randomly masks out the majority of the image patches and then reconstructs them, which has achieved great success on downstream tasks. Inspired by MAE, many works (Tong et al., 2022; Bachmann et al., 2022) have begun to apply masked image prediction to diverse scenarios. Considering the strong representation ability of visual foundation models, we attempt to embed the density map to guide the masked prediction for intra-frame, performing density-driven regression from image to density map and forming an efficient self-representation learning framework for video object counting.

## 3 DRONEBIRD DATASET

Video object counting methods not only hold promising application prospects in human-centric activity analysis, but also possess invaluable potential in natural scenarios, such as migratory bird

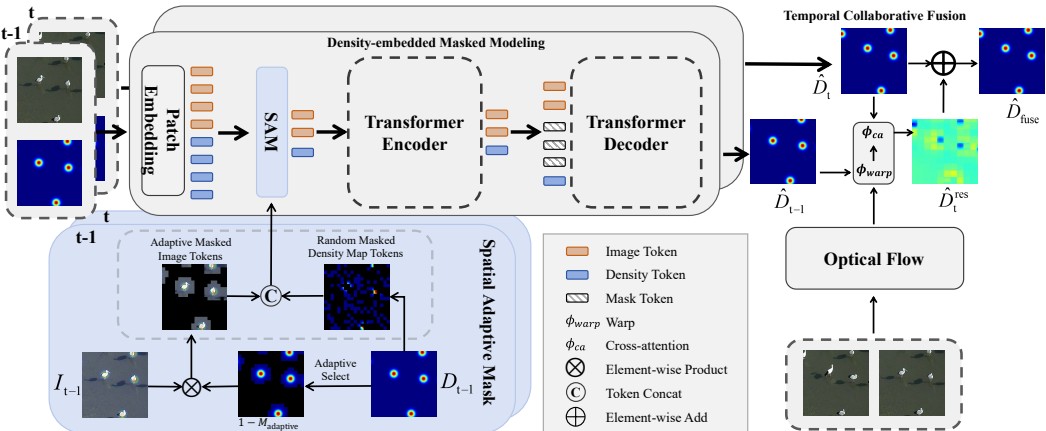

Figure 2: An overview of our E-MAC. For the temporal collaborative fusion, we use optical flow to fuse multi-frame density maps. For the density-embedded masked modeling, the image and density map are treated as multi-modal data and are fed into the transformer autoencoder for self-representation masked modeling simultaneously. The spatial adaptive masking uses the density map to balance the dynamic fore-background. During inference, the density map is fully masked.

protection. In the scenario of counting volant species like birds, to the best of our knowledge, the existing open-source data is largely limited to discrete image data (Arteta et al., 2016; Wang et al., 2023), which makes it challenging to apply these methods to dynamic bird activity analysis scenarios. To alleviate the issue of data scarcity as well as to assist in migratory bird activity analysis, we collected a new large-scale video bird dataset called *DroneBird*. DroneBird provides point annotations for bird counting, and also provides additional trajectory annotations for further bird tracking. To the best of our knowledge, DroneBird is the first bird dataset captured in video from a drone's viewpoint and provides both point annotations and trajectory annotations.

We have collected statistics on various aspects of our DroneBird dataset and compared them with some existing datasets in Table 4. All the videos in DroneBird are recorded at 30 frames per second with resolutions of $2160 \times 4096$ or $2160 \times 3840$. Each frame contains between 8 and 673 annotated objects, averaging $171.5$ per frame. The dataset includes $3,686,409$ bird annotations and $9,389$ bird trajectories, ranging from 1 to 500 frames in length. To further investigate DroneBird, we have analyzed five main attributes of each sample, i.e., *Illumination*, *Density*, *Perspective*, *Distance*, and *Posture*. We present the distribution of these attributes and their correlation in Fig. 1. Each arc represents an attribute, and each chord connects between two arcs, indicating that there are images that possess both attributes represented by the two arcs. For each attribute, we provide two example images in the DroneBird dataset for reference. Detailed descriptions of these attributes and clearer visualization are presented in Appendix A.1.

## 4 METHOD

In this paper, we introduce an **E**fficient **M**asked **A**utoencoder **C**ounting (**E-MAC**) framework based on a self-representation foundation model for video object counting. The framework of our E-MAC is depicted in Fig. 2, which consists of temporal collaborative fusion (TCF), density-embedded masked modeling (DEMO), and spatial adaptive masking (SAM). We utilize optical flow to establish connections across multiple frames to capture inter-frame information. A temporal residual map is constructed by leveraging optical flow information between frames, which utilizes historical data to enhance the counting performance of the current frame. For intra-frame information, we employ density-embedded masked modeling (DEMO) and spatial adaptive masking (SAM) based on the self-representation foundation model to effectively balance the learning on foreground and background for more accurate density map estimation.

### 4.1 TEMPORAL COLLABORATIVE FUSION

The temporal collaborative fusion aims to integrate multiple frames for more accurate estimation. Given the frames at time $t$ and $t - 1$, each sample consists of a frame image and a density map. The

samples of two frames can be described as $\mathbf{S}_t = \{I_t, D_t\}$ and $\mathbf{S}_{t-1} = \{I_{t-1}, D_{t-1}\}$, which are then fed into the DEMO for density-embedded masked modeling. Different from most existing methods, we take the density map as an auxiliary modality corresponding to the image modality.

Specifically, for a sample $\mathbf{S_t} = \{I_t, D_t\}$, the patch embedding module patchifies and embeds both the image modality and density map modality into multi-modal tokens. The SAM removes specific patches from these multi-modal tokens before the transformer encoder. After passing through the encoder, the masked positions of the density map are filled with random mask tokens. The decoder then reconstructs the complete original density map based on the incomplete input information. In our framework, two temporally adjacent samples $\{\mathbf{S}_t, \mathbf{S}_{t-1}\}$ are simultaneously fed into the DEMO, where the aforementioned process is used to complete the reconstruction and generation of the predicted density maps $\{\hat{D}_t, \hat{D}_{t-1}\}$.

The reconstructed density maps $\{\hat{D}_{t-1}, \hat{D}_t\}$ are obtained by the output of DEMO. To align their spatial distributions, a pre-trained optical flow network (Sun et al., 2018) estimates the motion displacement, followed by a warp operation on $\hat{D}_{t-1}$, resulting in $\hat{D}_{t-1}^{\text{warp}}$. The cross-attention between $\hat{D}_{t-1}^{\text{warp}}$ and $\hat{D}_t$ then produces $\hat{D}_t^{\text{res}}$, representing the temporal density residuals of adjacent frames. $\hat{D}_t^{\text{res}}$ and $\hat{D}_t$ are combined via element-wise addition to output the final fused prediction $\hat{D}_{\text{fuse}}$. The TCF can be formally described by Equation 1. The fusion effect is improved by utilizing an optical flow to align information between adjacent frames. We present the whole training process in Appendix A.2 to make it easy to understand.

$$\hat{D}_{\text{fuse}} = \underbrace{\left(\phi_{\text{ca}}(\phi_{\text{warp}}(\phi_{\text{OpticalFlow}}(I_t, I_{t-1}), \hat{D}_{t-1}), \hat{D}_t)\right)}_{\text{Temporal residual density of adjacent frames}} \oplus \hat{D}_t. \tag{1}$$

## 4.2 Density-embedded Masked Modeling

As depicted in Fig. 2, the density-embedded masked modeling (DEMO) is a Transformer-based autoencoder. The input sample $\mathbf{S}_t$ is first divided into patches $\{\mathbf{I}_{\text{patch}}, \mathbf{D}_{\text{patch}}\}$, which are then converted into a token sequence $\mathbf{T} \in \mathbb{R}^{B \times L \times C}$ where $B$ represents the batch size, $L$ is the number of tokens, and $C$ denotes the feature channels. The image $I_t$ and density map $D_t$ are tokenized simultaneously, then concatenated along the $L$ dimension, where $L = \mathcal{N}_I + \mathcal{N}_D$. $\mathcal{N}_I$ and $\mathcal{N}_D$ represent the number of tokens from the image and density map modalities. SAM is a density-guided masking strategy that uses human annotations as priors. Further details are provided in Sec.4.3. It retains $\mathcal{N}_I^{\text{ret}}$ foreground tokens from image $I_t$ and randomly keeps $\mathcal{N}_D^{\text{ret}}$ tokens from $D_t$, generating a new token sequence $\mathbf{T}^{\text{ret}} \in \mathbb{R}^{B \times (\mathcal{N}_I^{\text{ret}} + \mathcal{N}_D^{\text{ret}}) \times C}$.

The retained tokens are sent to the transformer encoder, while the remaining tokens are discarded and not passed into the Transformer. The output token dimension of the encoder is $B \times (\mathcal{N}_I^{\text{ret}} + \mathcal{N}_D^{\text{ret}}) \times D$. In the decoder, the retained density map tokens $\mathbf{T}_D^{\text{ret}}$ are separated from the retained token sequence $\mathbf{T}^{\text{ret}}$, where $\mathbf{T}^{\text{ret}} \in \mathbb{R}^{B \times (\mathcal{N}_I^{\text{ret}} + \mathcal{N}_D^{\text{ret}}) \times C}$ and $\mathbf{T}_D^{\text{ret}} \in \mathbb{R}^{B \times \mathcal{N}_D^{\text{ret}} \times C}$. The learnable random mask tokens are filled at the masked positions in the retained density map tokens $\mathbf{T}_D^{\text{ret}}$ as placeholders, and we use $\hat{\mathbf{T}}_D$ to represent the filled density map tokens. Cross-attention is then applied, with $\hat{\mathbf{T}}_D$ as the query and $\mathbf{T}^{\text{ret}}$ as the key and value. Then, the reconstructed density map $\hat{D}_t$ is generated by the two-layer transformer, as the end of the self-representation masked modeling.

## 4.3 Spatial Adaptive Masking

The masked modeling approach discards a subset of tokens prior to the transformer encoder, utilizing the decoder to reconstruct the missing information. This process allows the model to capture the relationships between tokens. In the context of multi-modal masked modeling, it further enables the model to learn associations and interaction mechanisms across different modalities. A substantial body of research indicates that random masking strategies may introduce excessive redundant information due to the imbalanced fore-background, which is detrimental to the model's learning process. To this concern, we developed spatial adaptive masking (SAM) for efficient learning of the dynamic changing targets in videos. This strategy reduces redundant background optimization and focuses the model's attention on the image foreground, thereby improving the efficiency of self-representation learning.

For a video frame $I$, its density distribution $D$ serves as the standard for delimiting the foreground and the background. The lower-left part of Fig. 2 provides a detailed illustration of the SAM. The symmetric Dirichlet distribution (Bachmann et al., 2022) is used to determine the number of retained tokens for the image modality and density map modality when generating multi-modal masks, denoted as $\mathcal{N}_I^{\text{ret}}$ and $\mathcal{N}_D^{\text{ret}}$, respectively. We calculate the number of targets $\mathbf{V}_D^i$ in the $i$-th density map patch $\mathbf{D}_{\text{patch}}^i$ corresponding to each token, and $\mathbf{V}_D = \{\mathbf{V}_D^1, \mathbf{V}_D^2, \cdots \mathbf{V}_D^{\mathcal{N}_I}\}$, where $\mathbf{V}_D^i = \phi_{\text{sum}}(\mathbf{D}_{\text{patch}}^i)$. $\phi_{\text{sum}}$ represents the pixel-wise sum operation in each density patch (Zhang et al., 2016; Li et al., 2018), and the results represent the number of targets in the corresponding patch. To focus on the foreground, we sort the image tokens according to the number of targets $\mathbf{V}_D^i$ in the corresponding density map modality. While the foreground provides more valid information, the background should not be completely ignored. Therefore, we set a **b**ackground **r**etention **p**robability (**BRP**) $\mathcal{P}$ to introduce the background information, where BRP determines the sorting manner, in ascending order with a probability of $\mathcal{P}$ (focus on the background) or in descending order with a probability of $1 - \mathcal{P}$ (focus on the foreground). Detailed experiments of $\mathcal{P}$ are presented in the Sec. 5.4. The first $\mathcal{N}_I^{\text{ret}}$ tokens are retained to guide the masking of image $I$, preserving the foreground while discarding the background. Here, we denote $\mathcal{K}$ as the set of positions that should be kept, and $\mathbb{N}$ is a random variable that follows a Uniform distribution between 0 and 1, which is produced by a random number generator.

$$\mathcal{K} = \begin{cases} \texttt{argsort}_{des}(\phi_{\text{sum}}(\mathbf{D}_{\text{patch}}^i))\{1 : \mathcal{N}_I^{\text{ret}}\}, & \text{if} \quad \mathbb{N} \leq 1 - \mathcal{P}, \\ \texttt{argsort}_{asc}(\phi_{\text{sum}}(\mathbf{D}_{\text{patch}}^i))\{1 : \mathcal{N}_I^{\text{ret}}\}, & \text{otherwise.} \end{cases} \tag{2}$$

Based on this, we can obtain the spatial adaptive mask $M_{\text{adaptive}} = \{M_{\text{adaptive}}^i | 1 \leq i \leq \mathcal{N}_I\}$ for image $I$. For each token in position $i$ and its corresponding mask $M_{\text{adaptive}}^i$, we have

$$M_{\text{adaptive}}^i = \begin{cases} 0, & \text{if} \quad i \in \mathcal{K}, \\ 1, & \text{otherwise}, \end{cases} \tag{3}$$

where 0 represent keeping and 1 represent masking. We denote the retained tokens from the image $I$ as $\mathbf{T}_I^{\text{ret}} \in \mathbb{R}^{B \times \mathcal{N}_I^{\text{ret}} \times C}$, where $\mathbf{T}_I^{\text{ret}} = \mathbf{T}_I \otimes (1 - M_{\text{adaptive}})$ and $\mathbf{T}_I$ denotes all the tokens of image $I$. For the density token $\mathbf{T}_D$ corresponding to density map patch $\mathbf{D}_{\text{patch}}$, we generate a *random mask* $M_{\text{random}}$ to retain $\mathcal{N}_D^{\text{ret}}$ tokens as $\mathbf{T}_D^{\text{ret}} \in \mathbb{R}^{B \times \mathcal{N}_D^{\text{ret}} \times C}$. These retained tokens are then concatenated to $\mathbf{T}^{\text{ret}} \in \mathbb{R}^{B \times (\mathcal{N}_I^{\text{ret}} + \mathcal{N}_D^{\text{ret}}) \times C}$ and fed into the decoder for prediction.

During inference, the density maps tokens are fully masked and removed. Only the image tokens are fed into the trained network, which is then required to fully reconstruct the density maps. In other words, we set $\mathcal{N}_D^{\text{ret}} = 0$ and $\mathcal{N}_I^{\text{ret}} = \mathcal{N}_I$.

## 4.4 Loss Function

In this work, we minimize the Mean Square Error (MSE) to ensure that both multi-frame fused density map $\hat{D}_{\text{fuse}}$ and the single-frame predicted result $\hat{D}_t$ approach the ground-truth density map $D_t$. To simplify the optimization of the optical flow network, we apply an MSE loss between the warped $\hat{I}_{t-1}^{\text{warp}}$ and the original image $I_t$:

$$\mathcal{L}_{\text{MSE}} = \frac{1}{2hw} \sum_{i=1}^{h} \sum_{j=1}^{w} \left( \hat{E}_{i,j} - G_{i,j} \right)^2, \tag{4}$$

where $\hat{E}$ and $G$ represent the estimated vector and its ground truth. Specifically, $\hat{E}$ in $\mathcal{L}_{\text{fuse}}, \mathcal{L}_{\text{cur}}, \mathcal{L}_{\text{opt}}$ represents $\hat{D}_{fuse}, \hat{D}_t$ and $\hat{I}_{t-1}^{\text{warp}}$ respectively, and the corresponding $G$ represents $D_t, D_t$ and $I_t$. A Total Variations (TV) loss (Rudin et al., 1992) is introduced as a regular term to encourage spatial smoothness in $\hat{D}_{\text{fus}}$. TV loss can be expressed as:

$$\mathcal{L}_{\text{TV}} = \frac{1}{hw} \sum_{i=1}^{h} \sum_{j=1}^{w} \left[ \left( \hat{D}_{\text{fuse}}^{i,j} - \hat{D}_{\text{fuse}}^{i-1,j} \right)^2 + \left( \hat{D}_{\text{fuse}}^{i,j} - \hat{D}_{\text{fuse}}^{i,j-1} \right)^2 \right]. \tag{5}$$

The objective loss function can be expressed as follows, where $\lambda_1$ - $\lambda_4$ are hyperparameters.

$$\mathcal{L} = \lambda_1 \mathcal{L}_{\text{fuse}} + \lambda_2 \mathcal{L}_{\text{cur}} + \lambda_3 \mathcal{L}_{\text{opt}} + \lambda_4 \mathcal{L}_{\text{TV}}. \tag{6}$$

Table 1: Quantitative comparison between our proposed method and existing methods with metrics MAE and RMSE, lower metrics better. Further comparative results can be found in Sec. A.3.

| Method | Type | Mall | | FDST | | VSCrowd | | DroneBird | |
|---|---|---|---|---|---|---|---|---|---|
| | | MAE↓ | RMSE↓ | MAE↓ | RMSE↓ | MAE↓ | RMSE↓ | MAE↓ | RMSE↓ |
| MCNN (Zhang et al., 2016) | Image | - | - | 3.77 | 4.88 | 27.1 | 46.9 | 122.35 | 149.07 |
| CSRNet (Li et al., 2018) | Image | 2.46 | 4.70 | 2.56 | 3.12 | 13.8 | 21.1 | 66.11 | 79.33 |
| CAN (Liu et al., 2019) | Image | - | - | - | - | - | - | 70.21 | 92.15 |
| MAN (Lin et al., 2022) | Image | - | - | 2.79 | 4.21 | 8.3 | 10.4 | _39.11_ | _50.08_ |
| HMoDE (Du et al., 2023) | Image | 2.82 | 3.41 | 2.49 | 3.51 | 19.8 | 39.5 | 67.47 | 81.40 |
| PET (Liu et al., 2023) | Image | 1.89 | 2.46 | 1.73 | 2.27 | _6.6_ | 11.0 | 45.10 | 52.35 |
| Gramformer (Lin et al., 2024) | Image | 1.69 | 2.14 | 5.15 | 6.32 | 8.09 | 15.65 | 49.11 | 65.50 |
| EPF (Liu et al., 2020) | Video | - | - | 2.17 | 2.62 | 10.4 | 14.6 | 97.22 | 133.01 |
| PFTF (Avvenuti et al., 2022) | Video | 2.99 | 3.72 | 2.07 | 2.69 | - | - | 89.76 | 101.02 |
| GNANet (Li et al., 2022) | Video | - | - | 2.10 | 2.90 | 8.2 | **10.2** | - | - |
| FRVCC (Hou et al., 2023) | Video | _1.41_ | _1.79_ | 1.88 | 2.45 | - | - | - | - |
| STGN (Wu et al., 2023) | Video | 1.53 | 1.97 | _1.38_ | _1.82_ | 9.6 | 12.5 | 92.38 | 124.67 |
| Ours | Video | **1.35** | **1.76** | **1.29** | **1.69** | **6.0** | _10.3_ | **38.72** | **42.92** |

# 5 EXPERIMENTS

## 5.1 EXPERIMENTAL SETTINGS

**Datasets and Metrics.** We conduct experiments on our DroneBird dataset and three video object counting datasets: Fudan-ShanghaiTech (FDST) (Fang et al., 2019), Mall (Loy et al., 2013) and VSCrowd (Li et al., 2022) datasets. We use a fixed Gaussian kernel ($\sigma = 6$) to generate the ground-truth density map on these datasets. Following previous methods, we evaluate the counting performance by using Mean Absolute Error (MAE) and Root Mean Square Error (RMSE) for each frame in datasets. MAE measures accuracy as $\mathrm{MAE} = \frac{1}{n}\sum_{i=1}^{n}\left|D_i - \hat{D}_i\right|$, and RMSE measures robustness as $\mathrm{RMSE} = \sqrt{\frac{1}{n}\sum_{i=1}^{n}\left(D_i - \hat{D}_i\right)^2}$, where $n$ is the number of samples in datasets, $D_i$ and $\hat{D}_i$ represent the ground truth and predicted density maps of the $i$-th sample, respectively.

**Implementation Details.** For the backbone network, the optical flow network leverages the pre-trained PWCNet (Sun et al., 2018), while the pre-trained ViT-B from MultiMAE (Bachmann et al., 2022) is used as the encoder in E-MAC. During inference, the density maps $\{D_{t-1}, D_t\}$ are fully masked, leaving the video frames $\{I_{t-1}, I_t\}$ intact, enabling the model to reconstruct the complete density map $\hat{D}_{\mathrm{fus}}$ from the input video alone. Our experiments are conducted on Huawei Atlas 800 Training Server with CANN and NVIDIA RTX 3090 GPU. We adopt random horizontal flipping to perform data augmentation. Density maps are standardized by mean and standard deviation for better optimization. For hyperparameter settings, the model employs a linear learning rate warm-up for the first 15 epochs, followed by a cosine decay learning rate. The weight decay of AdamW is set to 0.05, and layer decay is set to 0.75 for the encoder. The mask ratio is 0.72. Empirically, to maintain a balance between foreground and background tokens, setting a small probability $\mathcal{P}$ of retaining only the background improves the model's performance for SAM. The probability $\mathcal{P}$ for spatial adaptive masking is set to 0.2. The trade-off parameters $\lambda_1, \lambda_2, \lambda_3, \lambda_4$ are set to 10, 10, 1, and 20, respectively.

## 5.2 COMPARISONS

We compare our method with several state-of-the-art methods on our DroneBird dataset, the FDST dataset, the Mall dataset, and the VSCrowd dataset.

**Mall.** Mall provides video data from a fixed viewpoint in a shopping mall, where factors such as lighting are relatively controllable. On the Mall dataset, we follow the previous works (Bai & Chan, 2021; Hossain et al., 2020) for a fair comparison. The model is trained with the first 800 frames of the Mall dataset, and the rest 1,200 frames are used as the test set. The input images are set to the size of $448 \times 640$ and the batch size is set to 3. The quantitative comparisons are reported in Table 1. Our method achieved significant advantages on MAE and RMSE metrics, which improves 4% of MAE and 2% of RMSE compared to the runner-up method FRVCC (Hou et al., 2023) based on CSRNet (Li et al., 2018). Compared to the PFTF, our method achieves significant reductions in MAE and RMSE of 55% and 53%. We compared our method to a video counting

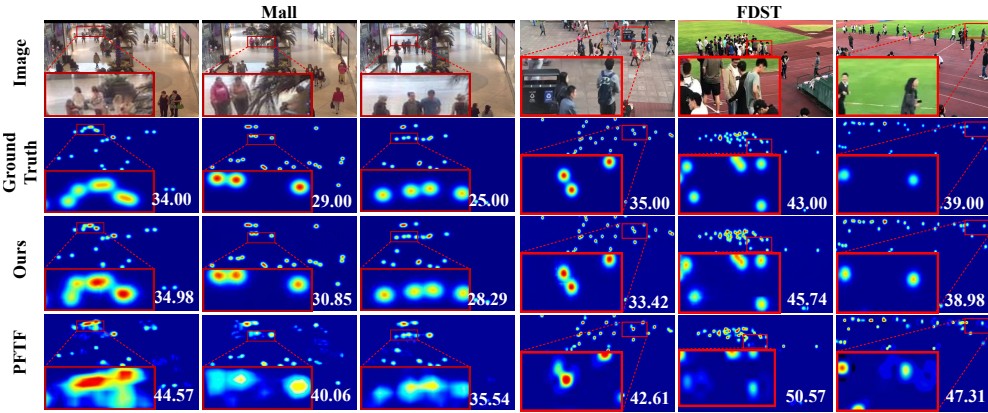

Figure 3: Visualized comparisons on the FDST dataset and the Mall dataset.

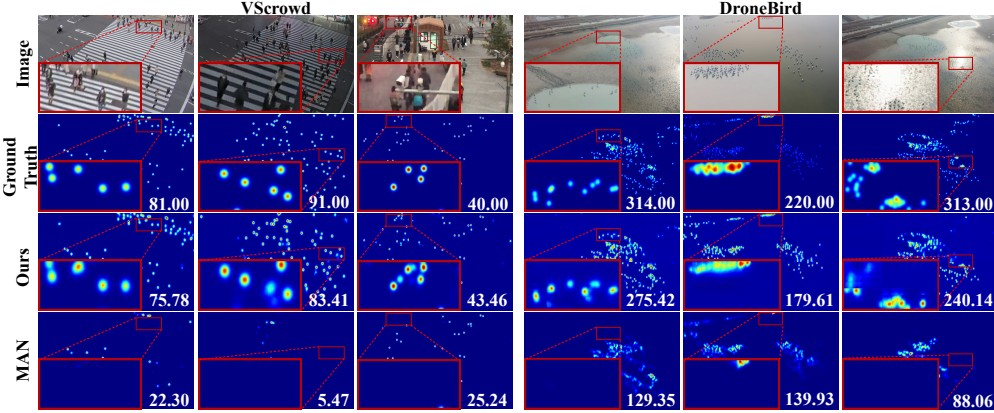

Figure 4: Visualized comparisons on the VScrowd dataset and our DroneBird dataset.

method PFTF (Avvenuti et al., 2022) and visualized the results on the Mall dataset in Fig. 3. Our method produces more clear and accurate density distributions of distant low-pixel targets, resulting in superior visualization performance. The quantitative and qualitative experimental results proved the superiority of our framework in the indoor scenarios.

**FDST.** The FDST dataset provides a wider range of scenarios, including various outdoor scenes, with more diverse variables compared to the Mall dataset, thus posing greater challenges. For quantitative comparison, we reported the MAE and RMSE metrics of our model and competing methods on the FDST dataset in Table 1. The result shows that our method achieves the best MAE and RMSE, decreasing the two metrics of 7% compared to the runner-up method STGN (Wu et al., 2023), and 31% compared to FRVCC (Hou et al., 2023), respectively. For qualitative comparison, we visualize the predicted results of our method and the competing video counting method PFTF (Avvenuti et al., 2022) on several scenarios in Fig. 3. Our method offers better visualization effects and delivers more accurate quantitative predictions. These experimental results validate our method maintains superior performance in more complex scenarios, such as outdoor environments.

**VSCrowd.** VSCrowd collected more videos by using surveillance cameras or the Internet. Compared to FDST, the VSCrowd dataset provides a more diverse and complex set of outdoor scenes and presents greater challenges for video crowd counting. The evaluation results of our method and the competing method on the VSCrowd dataset are presented in Table 1. Compared to existing methods, our approach achieves superior performance on MAE and runner-up performance on RMSE metrics. Compared to the recent video counting method STGN (Wu et al., 2023), our method improves the performance by 38% in MAE and 18% RMSE, respectively. Compared to the runner-up method GNANet (Li et al., 2022), our method beat GANNet on the MAE metric and achieved competitive performance on the RMSE metric. We present detailed visualizations on the VSCrowd datasets in Fig. 4. Our method achieves accurate counting results under low-light or long-distance dense conditions compared to the previous method (Lin et al., 2022), showing the priority of our framework. These quantitative and qualitative comparison results demonstrate that our framework still possesses competitive performance in more diverse and complex outdoor scenarios.

Table 2: Ablation studies on three components.

| Exp. | DEMO | SAM | TCF | MAE↓ | RMSE↓ |
|------|------|-----|-----|------|-------|
| I    |      |     |     | 2.45 | 3.22  |
| II   |      |     | ✓   | 2.32 | 2.69  |
| III  |      | ✓   | ✓   | 1.57 | 1.99  |
| IV   | ✓    |     | ✓   | 1.69 | 2.20  |
| V    | ✓    | ✓   | ✓   | **1.29** | **1.69** |

Table 3: Effect of each loss function.

| Exp. | $\mathcal{L}_{\text{fuse}}$ | $\mathcal{L}_{\text{cur}}$ | $\mathcal{L}_{\text{opt}}$ | $\mathcal{L}_{\text{TV}}$ | MAE↓ | RMSE↓ |
|------|------|------|------|------|------|-------|
| VI   | ✓    |      |      |      | 1.87 | 2.50  |
| VII  | ✓    |      |      | ✓    | 1.80 | 2.37  |
| VIII | ✓    |      | ✓    | ✓    | 1.60 | 2.03  |
| IX   | ✓    | ✓    |      | ✓    | 1.39 | 1.77  |
| X    | ✓    | ✓    | ✓    | ✓    | **1.29** | **1.69** |

**DroneBird.** Different from the previous three datasets, our DroneBird provides bird flock data from a drone's perspective, with scenes mostly consisting of open outdoor areas and exhibiting higher dynamics which pose significant challenges for video object counting. We assessed several existing methods on our dataset, as detailed in Table 1. Our method outperforms both recent video and image counting techniques, achieving a $58\%$ improvement in MAE and a $66\%$ improvement in RMSE compared to the STGN (Wu et al., 2023) method. Additionally, our approach shows enhancements in MAE and RMSE over the previous optimal method, MAN (Lin et al., 2022). For qualitative comparison, we compared the visualization results in multiple challenging scenarios. Our method achieves more accurate counting results of birds, even in complex areas like water reflections. The quantitative experimental results across different attributes and the visualization effects demonstrate that our framework still exhibits superior counting performance in even more complex and variable outdoor scenes from a drone's perspective, thereby highlighting the superiority of our framework. Furthermore, we conduct attribute comparisons with three competing methods (Avvenuti et al., 2022; Liu et al., 2020; Wu et al., 2023) on various attributes of the DroneBird dataset, as illustrated in Fig. 1. These experiments fully demonstrate our superiority in various complex scenarios.

## 5.3 ABLATION STUDY

We perform the ablation study on the FDST dataset to investigate the effectiveness of Density-embedded masked modeling (DEMO), spatial adaptive masking (SAM), and temporal collaborative fusion (TCF). We construct the same architecture as that in comparison experiments and trained for 200 epochs. The hyperparameters are set to the same as the previous experiments on the FDST dataset unless otherwise noted.

We test five variants of our method to assess the impact of DEMO, SAM, and TCF in experiments I-V, with results in Table 2. Exp.I is the E-MAC baseline, using a pure transformer for density map regression. Exp.II adds optimal flow and a fusion module for inter-frame relationships. Exp.III incorporates SAM into Exp.II for adaptive masking. Exp.IV introduces DEMO to Exp.II, masking both images and density maps, unlike Exp.III which only masks images with SAM. Exp.V combines DEMO and SAM to evaluate overall performance.

**Effect of `TCF`.** We incorporated the optical flow module and fusion module into Exp.II and compared its performance with Exp.I. The results indicate that the construction of inter-frame relationships brought a performance improvement of $5\%$ to $16\%$ in terms of MAE and RMSE. By employing optical flow mapping, we were able to effectively leverage the inherent temporal information present in video data, enhancing the information of the current frame and improving the overall performance. Further study and visualization on TCF are presented in Appendix A.5.

**Effect of `DEMO`.** As shown in Exp.II and Exp. IV in Table 2, `DEMO` brings in $27\%$ and $18\%$ improvement on MAE and RMSE metrics. In Exp.V, the introduction of the self-representation learning of density maps resulted in $17\%$ and $15\%$ improvement in MAE and RMSE compared to Exp.III. The self-representation learning of density maps implicitly drives the regression of density maps and effectively boosts the counting performance.

**Effect of `SAM`.** In Exp.III, foreground tokens from images are selected while all image tokens are selected in Exp.II. As shown in Exp.II and Exp.III, our proposed spatial adaptive masking brings an improvement of $32\%$ in MAE and $26\%$ in RMSE. Exp.II and Exp.III show the effect of our proposed `SAM`. Additionally, `SAM` brought $23\%$ performance improvement to `DEMO` in Exp.V. Hyperparameter $\mathcal{P}$ is set to 0.2 in Exp.III and Exp.IV.

## 5.4 DISCUSSION

**Loss Analysis.** We evaluate different loss functions in our framework (Table 3), training for 200 epochs. Starting with Exp.VI (baseline using $\mathcal{L}_{\text{fuse}}$), we incrementally add loss terms. Adding $\mathcal{L}_{\text{TV}}$

Figure 5: Hyperparameter analysis of background retention probability, mask ratio, and loss weights.

(Exp.VII) improves MAE and RMSE by $4\%$ and $5\%$. Introducing $\mathcal{L}_{opt}$ (Exp.VIII) brings further gains of $11\%$ and $14\%$ over Exp.VII. Exp.IX adds $\mathcal{L}_{cur}$, achieving $23\%$ and $25\%$ improvement in MAE and RMSE compared to Exp.VII. Exp.X combines $\mathcal{L}_{opt}$ and $\mathcal{L}_{cur}$, yielding $19\%$ and $17\%$ improvements over Exp.VII. The stronger impact of $\mathcal{L}_{cur}$ stems from its direct supervision for density map generation, outperforming other loss terms.

**Impact of Background Retention Probability.** We have conducted an in-depth analysis for our SAM. Considering that discarding background redundant information altogether leads to imbalanced learning towards foreground, which in turn exhibits a decrease in performance. On the other hand, omissions during manual annotation could result in some counting information being present in the background. Therefore, we retain only the background of $I_t$ during SAM with a certain probability $\mathcal{P}$. Based on Exp.V, we conduct further experiments on the choices of $\mathcal{P}$. We choose five sampling points: $0, 0.1, 0.2, 0.4$, and $1$, in our experiments and compared them with the Exp.V in the ablation study. The subfigure (a) of Fig. 5 provides a more intuitive view of the final performance with respect to the probability $\mathcal{P}$. The horizontal axis indicates the probability of sorting the tokens in ascending order. We notice that the curve shows a clear downward rebound trend, and the quantitative metrics show a decline of different degrees in both four experiments compared to Exp.V. We finally choose $0.2$ as the default probability in our experiments.

**Impact of Mask Ratio.** We conducted experiments to evaluate the impact of the mask ratio on `DEMO` by varying it from $0.67$ to $0.83$. To more clearly evaluate the impact of the mask ratio, these experiments were specifically performed on our E-MAC without considering temporal information. Fig. 5(b) shows the results. As the mask ratio decreases, more tokens are input, and MAE decreases. However, at a mask ratio of $0.67$, performance drops by $7\%$ compared to $0.72$. We conclude that a lower mask ratio provides sufficient information for reconstruction, but too low a ratio introduces redundant information, harming performance.

**Hyperparameter Analysis.** We conducted experiments on the setting of the hyperparameters $\{\lambda_1, \lambda_2, \lambda_3, \lambda_4\}$ of each loss function, as shown in subfigures (c-f) of Fig 5. We vary the weights of the remaining loss terms while fixing the weights of the other three loss terms, and the experimental results correspond to the four subfigures of Fig 5. We finally fixed the weights $\lambda_1, \lambda_2, \lambda_3, \lambda_4$ of each loss term to $10, 10, 1, 20$ in our experiments.

## 6    CONCLUSION

This paper aims to address the dynamic imbalance of the fore-background in video object counting. Considering the dynamic sparsity of foreground objects, we proposed a density-embedded efficient masked autoencoder counting framework. We introduced the self-representation foundation model to video object counting, which first takes the density map as an auxiliary modality and develops density-embedded masked modeling (`DEMO`) to drive the regression of density map estimation. To handle the infra-frame dynamic density distribution and make the model focus more on the foreground region in the self-representation learning, we proposed a simple but efficient **S**patial **A**daptive **M**asking (`SAM`), which dynamically generates masks depending on density maps to eliminate the effect of redundant background information and boost the performance. Furthermore, accounting for the inter-frame dynamism and utilizing the inherent temporal information in the video, we introduce the optical flow and propose a temporal collaborative fusion that learns to harness the inter-frame differences. Besides, we first proposed a new large-scale video bird dataset in the drone perspective, named DroneBird. Our DroneBird provides point and trajectory annotations in different scenes for counting and further localization and tracking tasks. Experimental results verify our superiority.

ACKNOWLEDGMENTS

This work was sponsored by the National Science and Technology Major Project (No. 2022ZD0116500), the National Natural Science Foundation of China (No.s 62476198, 62436002, U23B2049, 62222608, 62276186, 61925602), the Tianjin Natural Science Funds for Distinguished Young Scholar (No. 23JCJQJC00270), the Zhejiang Provincial Natural Science Foundation of China (No. LD24F020004). This work was also sponsored by CAAI-CANN OpenFund, developed on OpenI Community. The authors appreciate the suggestions from ICLR anonymous peer reviewers.

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

# A APPENDIX

## A.1 DRONEBIRD DATASET

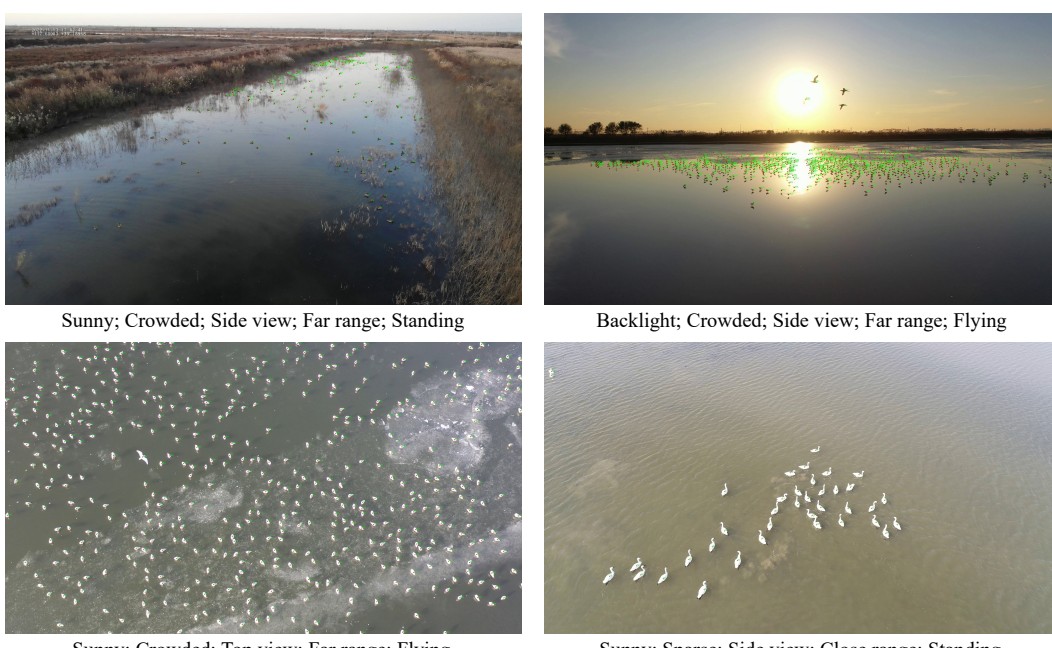

Sunny; Crowded; Side view; Far range; Standing

Backlight; Crowded; Side view; Far range; Flying

Sunny; Crowded; Top view; Far range; Flying

Sunny; Sparse; Side view; Close range; Standing

Figure 6: Visualization of partial examples of DroneBird.

Our DroneBird dataset is captured by cameras mounted on drones using consumer drones such as the DJI Mavic 2 Pro, Phantom 4 Pro, etc. DroneBird captures a wide range of scenarios, including rivers, wetlands, lakes, ice, and other common bird habitats. The data captured in DroneBird is primarily obtained by the drone from 30 meters or 60 meters in the air, with a small portion of the data captured close to the ground. DroneBird's collection time is during the daytime or in the early evening when the weather conditions are favorable, and the view is relatively clear.

Table 4: Existing bird datasets and our proposed DroneBird dataset.

| Dataset | Type | Trajectory | Highest Resolution | Frames | Ave count | Total count |
|---|---|---|---|---|---|---|
| Penguin (Arteta et al., 2016) | Image | × | $1536 \times 2048$ | $33,405$ | $178.4$ | $5,970,899$ |
| Bird-Count (Wang et al., 2023) | Image | × | $768 \times 1024$ | $1,372$ | $131.1$ | $173,458$ |
| **DroneBird** | Video | ✓ | $2160 \times 4096$ | $21,500$ | $171.5$ | $3,686,409$ |

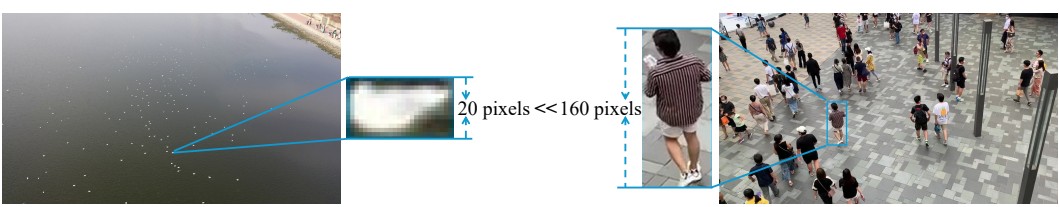

Figure 7: Visualization of pixel occupied by the target in DroneBird and existing crowd dataset.

DroneBird captured 50 videos of migratory birds and segmented them. Specifically, we used 30 of the video data as a $train$ set, 10 of the remaining 20 videos as a $test$ set, and 10 as a $validate$ set. We cut the 40 videos in the $train$ and $test$ sets to 500 frames

Table 5: Transfer performance on bird data.

| Exp. | Original dataset | Target dataset | MAE ↓ | RMSE ↓ |
|------|------------------|----------------|--------|--------|
| 1 | VSCrowd | DroneBird | 183.31 | 217.42 |
| 2 | DroneBird | DroneBird | 38.72 | 42.92 |

per video (around 17s), and cut the 10 videos in the $validate$ set to 150 frames per video (around 5s) to accomplish a reasonable data division. The $train$ set, $test$ set and $validate$ set after the division is completed contain $15,000$ frames, $5,000$ frames and $1,500$ frames, respectively. It is worth noting that the data scenarios for each of the two divisions in the $train$, $test$ and $validate$ sets are different, as a way to ensure that the data will not be leaked during the training process.

We have compiled the dataset's statistics and compared them with existing bird datasets, with the results shown in Table 4. A few examples of DroneBird are demonstrated in Fig. 6. Most of the targets in DroneBird are small in size. We compared the pixel height occupied by individual bird targets in DroneBird with that occupied by individuals in existing crowd data, and visualized this comparison in Fig. 7. The pixel height occupied by individual bird targets is significantly smaller than that of crowd individuals, posing a significant challenge for the target counting task.

We summarize the contributions and new challenges brought by DroneBird into the following four points:

- **Small target**: We compared the number of foreground pixels occupied by independent targets in the DroneBird data and the existing crowd data at the same resolution, and the independent targets in DroneBird occupy fewer pixels (less than 20 pixels height) than the independent targets in the existing dataset (more than 160 pixels height). Note that although the counting target in the crowd counting task is the human head, the human body still provides important information, and the human body occupies more pixels compared to birds, so our dataset provides data on small targets for counting. On the other hand, to avoid drone interference with bird activity, DroneBird's drone was photographed farther away from the flock, resulting in a further reduction of pixels occupied by the photographed individual birds.

- **Sparse distribution**: Unlike humans, birds in nature need enough space to move around, which means birds are more sparsely distributed in space. Coupled with the smaller target size of birds, the sparseness of the foreground portion is even more pronounced in DroneBird compared to the already existing dataset.

- **Fast movement**: Unlike existing video crowd-sourced data, the unique flight movements of birds allow them to do fast movements, posing a challenge for better temporal modeling.

- **New counting category**: DroneBird fills the gap of missing video bird data in the field. The models trained on crowd data fail to conduct bird counting directly (See Table 5). We evaluated the performance of our method trained on the VSCrowd dataset (Exp. 1) and the DroneBird training set (Exp. 2) on the DroneBird testing set. The results show that the model trained on the VSCrowd dataset failed to perform well on the bird data, which indicates the large gap between crowd and bird data.

## A.2 DETAILED METHOD

We detail the training process of our E-MAC method in Algorithm 1. The training process of DEMO is represented in Algorithm 2.

As shown in Algorithm 1, our model requires two frames $(I_t, I_{t-1})$ and their corresponding density maps $(D_t, D_{t-1})$ as input. The image and density map pairs $\mathbf{S_t} = \{I_t, D_t\}$ and $\mathbf{S_{t-1}} = \{I_{t-1}, D_{t-1}\}$ are fed to the DEMO to predict the density map $\hat{D}_t$ and $\hat{D}_{t-1}$. Meanwhile, a pretrained optical flow estimation network (Sun et al., 2018) is performed on $\hat{D}_t$ and $\hat{D}_{t-1}$ to generate optical flow $\mathcal{M}$, which is used to *warp* the $\hat{D}_{t-1}$ to $\hat{D}_{t-1}^{\text{warp}}$. Then the $\hat{D}_{t-1}^{\text{warp}}$ and $\hat{D}_t$ are fed into a cross-attention layer to calculate the residual $\hat{D}_t^{\text{res}}$ of adjacent predicted density map. The final predicted density map $\hat{D}_{\text{fuse}}$ of $I_t$ is then calculated by a pixel-wise add of $\hat{D}_t$ and $\hat{D}_t^{\text{res}}$.

---

**Algorithm 1** Framework Workflow in Training Phase

---

**Ensure:** $\{I_t, D_t\}, \{I_{t-1}, D_{t-1}\}$
**Require:** $\hat{D}_{\text{fuse}}$
1: **for all** epoch **do**
2: $\quad \hat{D}_t \leftarrow \phi_{\text{DEMO}}(I_t, D_t)$
3: $\quad \hat{D}_{t-1} \leftarrow \phi_{\text{DEMO}}(I_{t-1}, D_{t-1})$
4: $\quad \mathcal{M} \leftarrow \phi_{\text{OpticalFlow}}(I_t, I_{t-1})$
5: $\quad \hat{D}_{t-1}^{\text{warp}} \leftarrow \phi_{\text{warp}}(\mathcal{M}, \hat{D}_{t-1})$
6: $\quad \hat{D}_t^{\text{res}} \leftarrow \phi_{\text{ca}}(\hat{D}_{t-1}^{\text{warp}}, \hat{D}_t)$
7: $\quad \hat{D}_{\text{fuse}} \leftarrow \hat{D}_t + \hat{D}_t^{\text{res}}$
8: $\quad \mathcal{L}_{\text{fuse}}, \mathcal{L}_{\text{cur}}, \mathcal{L}_{\text{opt}}, \mathcal{L}_{\text{TV}} \leftarrow \phi_{\text{Loss}}$
9: $\quad \mathcal{L} \leftarrow \lambda_1 \mathcal{L}_{\text{fuse}} + \lambda_2 \mathcal{L}_{\text{cur}} + \lambda_3 \mathcal{L}_{\text{opt}} + \lambda_4 \mathcal{L}_{\text{TV}}$
10: **end for**

---

---

**Algorithm 2** DEMO workflow in training phase

---

**Ensure:** $I, D$
**Require:** $\hat{D}$
1: $\mathbf{T}_I, \mathbf{T}_D, \mathbf{I}_{\text{patch}}, \mathbf{D}_{\text{patch}} \leftarrow \phi_{\text{patchify}}(I, D)$
2: $\mathbf{V}_D \leftarrow \phi_{\text{sum}}(\mathbf{D}_{\text{patch}})$
3: $\mathbb{N} = \texttt{random}(0, 1)$
4: **if** $\mathbb{N} \leq 1 - \mathcal{P}$ **then**
5: $\quad \mathcal{K} \leftarrow \texttt{argsort}_{des}(\mathbf{V}_D)$
6: **else**
7: $\quad \mathcal{K} \leftarrow \texttt{argsort}_{asc}(\mathbf{V}_D)$
8: **end if**
9: **for all** $t_i \in \mathbf{T}_I$ **do**
10: $\quad$ **if** $i \in \mathcal{K}$ **then**
11: $\quad\quad M^i \leftarrow 0$
12: $\quad$ **else**
13: $\quad\quad M^i \leftarrow 1$
14: $\quad$ **end if**
15: **end for**
16: $\mathbf{T}_I^{\text{ret}}, \mathbf{T}_D^{\text{ret}} \leftarrow \texttt{mask}(\mathbf{T}_I, M_{\text{adaptive}}), \texttt{mask}(\mathbf{T}_D, M_{\text{random}})$
17: $\mathbf{T}^{\text{ret}} \leftarrow \phi_{\text{concate}}(\mathbf{T}_I^{\text{ret}}, \mathbf{T}_D^{\text{ret}})$
18: $\mathbf{T}^{\text{ret}} \leftarrow \phi_{\text{encoder}}(\mathbf{T}^{\text{ret}})$
19: $\mathbf{T}_I^{\text{ret}}, \mathbf{T}_D^{\text{ret}} \leftarrow \texttt{split}(\mathbf{T}^{\text{ret}})$
20: $\text{Mask} = \texttt{random}$
21: $\hat{\mathbf{T}}_D \leftarrow \phi_{\text{fill}}(\mathbf{T}_D^{\text{ret}}, \text{Mask})$
22: $\hat{\mathbf{T}}_D \leftarrow \phi_{\text{ca}}(\hat{\mathbf{T}}_D, \mathbf{T}^{\text{ret}})$
23: $\hat{D} \leftarrow \phi_{\text{decoder}}(\hat{\mathbf{T}}_D)$

---

In the DEMO, a video frame $I$ and its corresponding density map $D$ performed a patchify and a projection operation to get the token set $\mathbf{T}_I, \mathbf{T}_D$ and patch set $\mathbf{I}_{\text{patch}}, \mathbf{D}_{\text{patch}}$. Then we perform mask operation to get the retained tokens $\mathbf{T}_I^{\text{ret}}, \mathbf{T}_D^{\text{ret}}$. The number of retained tokens for the image and density map modalities are generated by the Dirichlet distribution, denoted as $\mathcal{N}_I^{\text{ret}}$ and $\mathcal{N}_D^{\text{ret}}$, respectively. Specifically, for image modality, we use an adaptive mask to obtain $\mathbf{T}_I^{\text{ret}}$ from $\mathbf{T}_I$. Firstly, we calculate the number of targets $\mathbf{V}_D^i$ in the $i$-th density map patch $\mathbf{D}_{\text{patch}}^i$, and $\mathbf{V}_D^i = \phi_{\text{sum}}(\mathbf{D}_{\text{patch}}^i)$, where $\phi_{\text{sum}}$ represents the pixel-wise sum operation in each density map patch, and the result represents the number of targets in the corresponding patch. To focus on the foreground, we sort the image tokens according to the number of targets $\mathbf{V}_D^i$ in the corresponding density map patch. Although the foreground provides more valid information, the background should not be completely ignored. Therefore, to introduce background information, we set a background retention probability (BRP) $\mathcal{P}$ to control the sorting manner. The tokens are sorted in ascending order with a probability of $\mathcal{P}$ (focus on background) or in descending order with a probability of $1 - \mathcal{P}$ (focus on

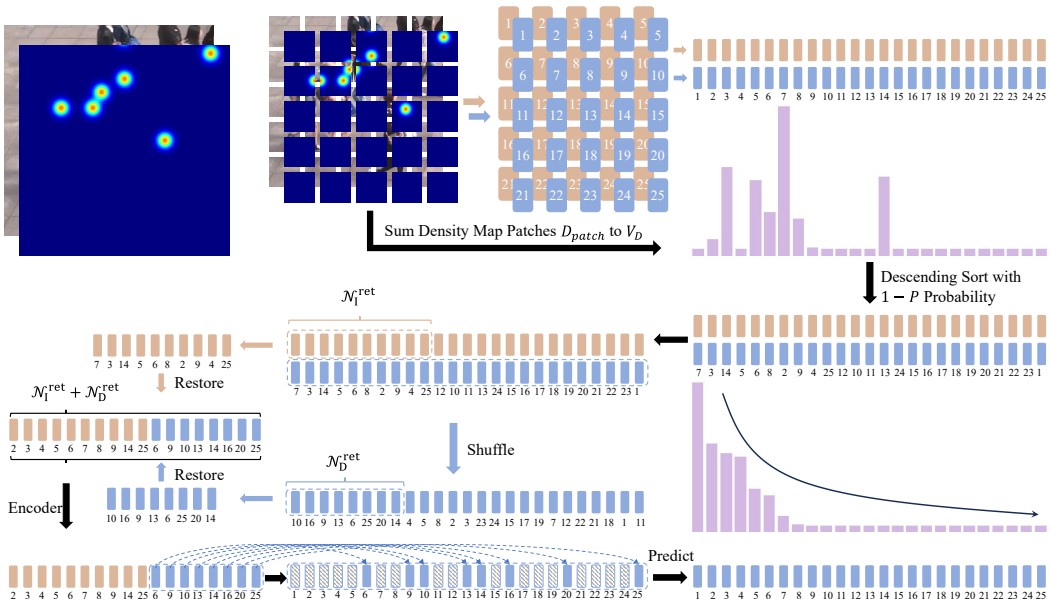

Figure 8: The detailed process of SAM. For ease of expression, we crop the image and the density map into 25 patches, as shown in the figure, this number varies according to the size of input images (each patch is set to $16 \times 16$). We first calculate the $\mathbf{V}_D^i$ of each density map patch $\mathbf{D}_{patch}^i$, and sort the token according to the number of targets $\mathbf{V}_D^i$. To balance the fore- and back-ground information, we set a background retention probability (BRP) $\mathcal{P}$ to determine the sorting manner, which is detailed in Sec. 4.3. For tokens from image modality, we keep the first $\mathcal{N}_I^{\text{ret}}$ tokens after sorting. For tokens from density map modality, we randomly **shuffle** their order and keep the first $\mathcal{N}_I^{\text{ret}}$ tokens, i.e., randomly select $\mathcal{N}_I^{\text{ret}}$ density tokens. Since the masked tokens are filled by learnable tokens in the decoder, we first restore the retained image and density tokens to their original order before sorting. Then, we concatenate and feed them into the encoder. Note that the restoration follows the same setting with Bachmann et al. (2022), which can be performed in the decoder as well. The full set of retained tokens and filled density map tokens are then fed into the decoder to predict the density map. Specifically, the full set of retained tokens are treated as *key* and *value* vectors, and the filled density map tokens are treated as *query* vector in the cross-attention layer in the decoder. Then, the output of the cross-attention layer is fed into two self-attention layers to perform the final prediction.

foreground). The first $\mathcal{N}_I^{ret}$ tokens after sorting is then retained as $\mathbf{T}_I^{\text{ret}}$. For density map modality, we randomly shuffle the order of $\mathbf{T}_D$ and retain the first $\mathcal{N}_D^{\text{ret}}$ tokens (randomly mask) as $\mathbf{T}_D^{\text{ret}}$. The retained tokens $\mathbf{T}_I^{\text{ret}}$ and $\mathbf{T}_D^{\text{ret}}$ are jointly fed into the encoder, while the remaining tokens are discarded and not passed into the model. The output tokens $\mathbf{T}^{\text{ref}}$ of encoder are then split to $\mathbf{T}_D^{\text{ret}}$ and $\mathbf{T}_I^{\text{ret}}$. The learnable random mask tokens are filled at the masked positions in $\mathbf{T}_D^{\text{ret}}$ as placeholders, and the filled tokens set is denoted as $\hat{\mathbf{T}}_D$. $\hat{\mathbf{T}}_D$ and $\mathbf{T}^{\text{ref}}$ are then fed in a cross-attention layer, in which $\hat{\mathbf{T}}_D$ servers as *query* and $\mathbf{T}^{\text{ref}}$ servers as *key* and *value*. Then, the reconstructed density map $\hat{D}$ is regressed by two transformer layers.

We have illustrated the detailed SAM process in Fig. 8. For ease of expression, we crop the image and density map into 25 patches. Each patch from the image and the density map at the same position is paired, and we have noted an index for each pair of patches in Fig. 8. For each pair of patches ($\mathbf{I}_{\text{patch}}^i$, $\mathbf{D}_{\text{patch}}^i$), we calculate the sum of the density map patch $\mathbf{V}_D^i$, sort the token pair ($\mathbf{T}_I$, $\mathbf{T}_D$) according to the number of targets $\mathbf{V}_D^i$. We set a background retention probability (BRP) $\mathcal{P}$ to determine the sorting manner, which is detailed in Sec. 4.3. For tokens from image modality, we retain the first $\mathcal{N}_I^{\text{ret}}$ tokens in the sorted $\mathbf{T}_I$. For tokens from density map modality, we randomly **shuffle** their order and retain the first $\mathcal{N}_D^{\text{ret}}$ tokens. Since the masked tokens will be filled by learnable tokens in the decoder, we first restore the retained image and density tokens to their original order

Table 6: Quantitative comparison between the proposed method and existing methods on the Mall dataset with metrics MAE and RMSE.

| Method | Type | MAE↓ | RMSE↓ |
|---|---|---|---|
| CSRNet (Li et al., 2018) | Image | 2.46 | 4.70 |
| RPNet (Yang et al., 2020) | Image | 2.20 | 3.60 |
| TAN (Wu et al., 2020) | Image | 2.03 | 2.60 |
| HMoDE (Du et al., 2023) | Image | 2.82 | 3.41 |
| PET (Liu et al., 2023) | Image | 1.89 | 2.46 |
| Gramformer (Lin et al., 2024) | Image | 1.69 | 2.14 |
| ConvLSTM (Xiong et al., 2017) | Video | 2.24 | 8.50 |
| LSTN (Fang et al., 2019) | Video | 2.00 | 2.50 |
| E3D (Zou et al., 2019) | Video | 1.64 | 2.13 |
| MLSTN (Fang et al., 2020) | Video | 1.80 | 2.42 |
| MOPN (Hossain et al., 2020) | Video | 1.78 | 2.25 |
| Monet (Bai & Chan, 2021) | Video | 1.54 | 2.02 |
| PFTF (Avvenuti et al., 2022) | Video | 2.99 | 3.72 |
| FRVCC (Hou et al., 2023) | Video | _1.41_ | _1.79_ |
| STGN (Wu et al., 2023) | Video | 1.53 | 1.97 |
| Ours | Video | **1.35** | **1.76** |

Table 7: Quantitative comparison between our proposed method and existing methods on the FDST dataset with metrics MAE and RMSE, lower metrics better.

| Method | Type | MAE↓ | RMSE↓ |
|---|---|---|---|
| MCNN (Zhang et al., 2016) | Image | 3.77 | 4.88 |
| CSRNet (Li et al., 2018) | Image | 2.56 | 3.12 |
| ChfL (Shu et al., 2022) | Image | 3.33 | 4.38 |
| MAN (Lin et al., 2022) | Image | 2.79 | 4.21 |
| HMoDE (Du et al., 2023) | Image | 2.49 | 3.51 |
| PET (Liu et al., 2023) | Image | 1.73 | 2.27 |
| Gramformer (Lin et al., 2024) | Image | 5.15 | 6.32 |
| ConvLSTM (Xiong et al., 2017) | Video | 4.48 | 5.82 |
| LSTN (Fang et al., 2019) | Video | 3.35 | 4.45 |
| MLSTN Fang et al. (2020) | Video | 2.35 | 3.02 |
| EPF (Liu et al., 2020) | Video | 2.17 | 2.62 |
| MOPN (Hossain et al., 2020) | Video | 1.76 | 2.25 |
| PHNet Meng et al. (2021) | Video | 1.65 | 2.16 |
| GNANet (Li et al., 2022) | Video | 2.10 | 2.90 |
| PFTF (Avvenuti et al., 2022) | Video | 2.07 | 2.69 |
| FRVCC (Hou et al., 2023) | Video | 1.88 | 2.45 |
| STGN (Wu et al., 2023) | Video | _1.38_ | _1.82_ |
| Ours | Video | **1.29** | **1.69** |

before sorting. Then, we concatenate and feed them into the encoder. The full set of retained tokens and filled density map tokens are then fed into the decoder to predict the density map. Specifically, the full set of retained tokens are treated as $key$ and $value$ vectors, and the filled density map tokens are treated as $query$ vector in the cross-attention layer in the decoder. Then, the output of the cross-attention layer is fed into two self-attention layers to perform the final prediction.

## A.3 MORE EXPERIMENT RESULTS

Additional results on the Mall, FDST, and VSCrowd datasets are provided in Tabs. 6, 7, and 8. In an extensive survey, our method consistently achieved competitive results.

Table 8: Quantitative comparison between our proposed method and existing methods on the VSCrowd dataset with metrics MAE and RMSE, lower metrics better.

| Method | Type | MAE↓ | RMSE↓ |
|---|---|---|---|
| MCNN (Zhang et al., 2016) | Image | 27.1 | 46.9 |
| CSRNet (Li et al., 2018) | Image | 13.8 | 21.1 |
| Bayesian (Ma et al., 2019) | Image | 8.7 | 11.8 |
| MAN (Lin et al., 2022) | Image | 8.3 | 10.4 |
| HMoDE (Du et al., 2023) | Image | 19.8 | 39.5 |
| PET (Liu et al., 2023) | Image | 6.6 | 11.0 |
| Gramformer (Lin et al., 2024) | Image | 8.09 | 15.65 |
| EPF (Liu et al., 2020) | Video | 10.4 | 14.6 |
| GNANet (Li et al., 2022) | Video | 8.2 | **10.2** |
| STGN (Wu et al., 2023) | Video | 9.6 | 12.5 |
| Ours | Video | **6.0** | 10.3 |

Table 9: Effect of TCF.

| Exp. | Model | MAE↓ | RMSE↓ |
|---|---|---|---|
| 3 | ViT | 2.45 | 3.22 |
| 4 | ViT w/ TCF | 2.32 | 2.69 |
| 5 | E-MAC | 1.51 | 2.03 |
| 6 | E-MAC TCF | **1.29** | **1.69** |

Table 10: Effect of E-MAC architecture.

| Exp. | Model | MAE↓ | RMSE↓ |
|---|---|---|---|
| 7 | vanilla ViT | 2.63 | 3.31 |
| 8 | MultiMAE | 2.45 | 3.22 |
| 9 | E-MAC w/ vanilla ViT weight | 1.49 | 1.93 |
| 10 | E-MAC w/ MultiMAE weight | 1.29 | 1.69 |

## A.4 IMPACT OF IMAGE SIZE

We conducted experiments on the FDST dataset to explore the effect of image size on the performance of our E-MAC module. We tried a variety of image input sizes and compared their experimental results to validate the effect of input image size. When we increased the image size used for training from $224 \times 224$ to $480 \times 480$, the final MAE showed a downward trend in the figure and decreased from $1.93$ to $1.47$, which improved by $24\%$. However, the computational cost of the model increases exponentially as the size of the image increases (Dosovitskiy et al., 2020). Integrating temporal information further increases the computational cost, thereby affecting the overall performance. Therefore, we set the input image size to $320 \times 320$ on the FDST dataset, balancing the performance and computational cost.

## A.5 IMPACT OF TCF

Additionally, we have conducted additional ablation studies on the TCF module, as shown in Table 9. We compare the performance of the original ViT architecture (Exp. 3 in Table 9), the ViT framework augmented with TCF (Exp. 4 in Table 9), and E-MAC with or without TCF (Exp. 5 and 6 in Table 9). The results demonstrate that TCF provides performance improvements of $5\%$ and $15\%$ on the two architectures, respectively. To further demonstrate how the fusion module works, we visualized the input $\hat{D}_t$, and output $\hat{D}_{\text{fuse}}$ of the fusion module as well as the intermediate variable $\hat{D}_t^{\text{res}}$ under three datasets, as shown in Fig 9, we provide the original image $I_t$ and corresponding ground truth $D_t$ in the first two rows of the figure for reference. Compared with $\hat{D}_t$, the fusion module can well realize the correction of the current density map by correlating the residuals obtained from the previous and current predicted density map, which makes the integration of the fused density map $\hat{D}_{\text{fuse}}$ closer to the ground truth. We zoom in on some areas, and we notice that the fusion module can remove some of the background interference by correlating the front and back predicted density maps, which makes the final predicted density map better.

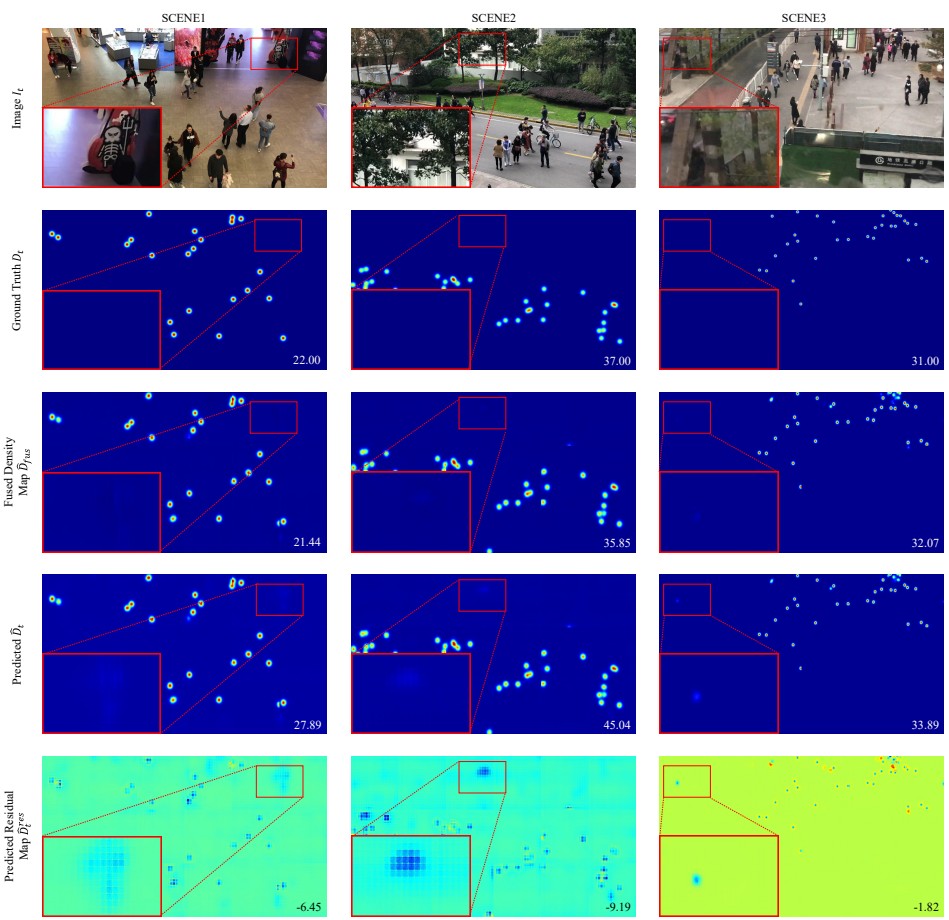

Figure 9: Visualization of the output and intermediate variables in the Fusion Module.

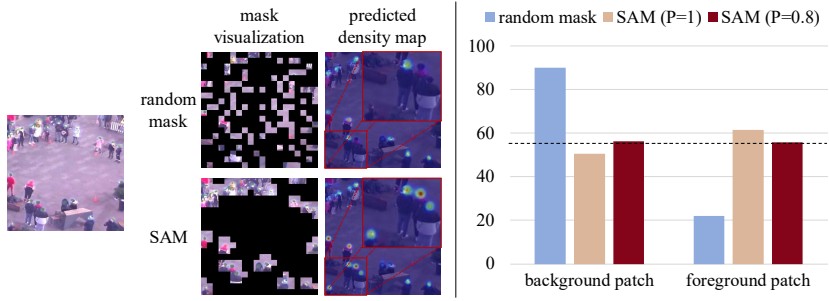

Figure 10: Visualization of the predicted density map w/ and w/o SAM, and statistical results of the number of foreground and background patches w/ and w/o SAM.

## A.6 VISUALIZATION OF SAM

We visualized the differences between the density maps predicted from the same image *w/* and *w/o* SAM and the ground truth, as presented in Fig. 10. The zoom-in regions display intuitive and significant visual differences. E-MAC trained *w/* SAM effectively counts the targets, while E-MAC using random masking fails to count some targets. Besides, we conducted a statistical comparison of the number of foreground and background patches in the images after applying random masks and different $\mathcal{P}$ (BRP) settings with SAM, as presented in Fig. 10. Obviously, SAM significantly reduces the proportion of background regions, thereby balancing the positive and negative samples.

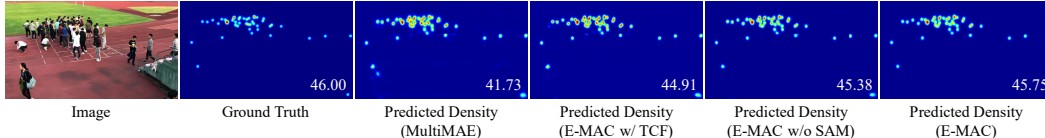

Figure 11: Visualization of the predicted results of our key components (`TCF`, `DEMO`, `SAM`).

Table 12: Comparison on FLOPs, FPS, and the number of Parameters.

| Method | FLOPs(G) ↓ | FPS ↑ | Parameters(M) ↓ | MAE (DroneBird) ↓ | RMSE (DroneBird) ↓ |
|---|---|---|---|---|---|
| EPF (Liu et al., 2020) | 815 | 19 | 20 | 97.22 | 133.01 |
| PFTF (Avvenuti et al., 2022) | 1075 | 13 | 23 | 89.76 | 101.02 |
| STGN (Wu et al., 2023) | 742 | 30 | 13 | 92.38 | 124.67 |
| Ours | 811 | 16 | 98 | 38.72 | 42,92 |

We have also visualized some of the predicted results of our key components (`TCF`, `DEMO`, `SAM`) with our baseline, as shown in Fig. 11. We gradually add the components (`TCF`, `DEMO`, `SAM`) to the baseline, resulting in increasingly accurate predictions and better visual results.

Table 11: Result with different random seeds.

| Metrics | 1 | 2 | 3 | Average |
|---|---|---|---|---|
| MAE | 1.29 | 1.28 | 1.33 | $1.30_{\pm 0.022}$ |
| RMSE | 1.69 | 1.66 | 1.69 | $1.68_{\pm 0.015}$ |

## A.7 EFFECT OF E-MAC ARCHITECTURE.

We provide additional quantitative experiments to demonstrate the performance sources of our proposed framework. As shown in Table 10, we provide a comparison of the performance of the vanilla ViT architecture using different pre-training weights (Exp. 7 and 8 in Table 10) as well as our framework using different pre-training weights (Exp. 9 and 10 in Table 10). As can be seen from the table: using a larger pre-training weight on ViT does provide better performance, but the improvement is very limited ($6\%$ improvement between Exp. 7 and 8). In contrast, our unique design tailored to the data and video counting tasks significantly enhances performance ($43\%$ between Exp. 7 and 9 and $47\%$ between Exp. 8 and 10 respectively).

## A.8 IMPACT OF RANDOM SEED.

To explore the impact of different random seeds on the overall performance of our model, we have also conducted more experiments on the FDST dataset, which is a relatively small dataset for quick validation. The average MAE and RMSE scores are $1.30_{\pm 0.022}$ and $1.68_{\pm 0.015}$. The results in Table 11 show that different random seeds do not significantly affect the performance of our model.

## A.9 MODEL EFFICIENCY DISCUSSION

We have compared FLOPs, FPS, and the number of parameters with some other video counting models and reported the results in Table 12. All the experiments were conducted on an NVIDIA RTX 3090 GPU. It is worth noting that the FLOPs and FPS of our method and the competing methods are comparable, although we first adopted the vision foundation model. Compared to EPF (Liu et al., 2020) and PFTF (Avvenuti et al., 2022), our E-MAC achieves superior performance with less required computations. STGN (Wu et al., 2023) achieves higher FPS with fewer parameters, but its performance is limited. Compared to STGN (Wu et al., 2023), our method achieved over 58% performance improvement with only 9% more FLOPs.

## A.10 DYNAMICS OF DATA

Here, we explore the intra-frame dynamics of the foreground regions in video data. We processed data in the FDST dataset and made analyses. We first utilized a $60 \times 60$ window to crop the images into patches and then counted the number of people in each patch.

Statistical result is shown in Fig. 12, the horizontal axis coordinates are the density of people in each patch, calculated from the corresponding density map. The left vertical coordinate is the number of patches for each crowd density. We fold a portion of the axis as the number gap is too large. The heterogeneous density distribution across image patches indicates inherent dynamism in the intra-frame density characteristics. Additionally, due to the presence of large background areas, the model should focus more on the foreground regions of the samples to extract the most informative and relevant information. The utilization of a fixed focus region across different samples may result in the loss of critical information about the foreground areas. Similarly, the adoption of completely random focus regions is unable to consistently capture the salient information within the foreground regions. Thus, we suggest the employment of a dynamic masking mechanism to obtain the foreground focus regions for different samples.

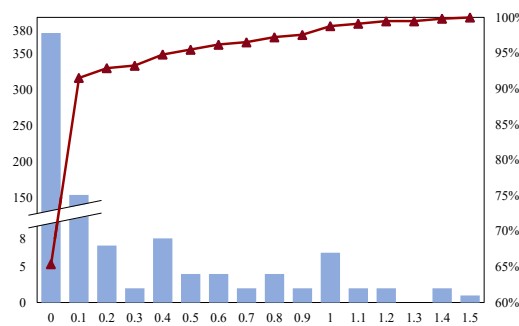

Figure 12: Density distribution. The bar graph portion (blue) represents the number of patches corresponding to the crowd density. The line graph portion (red) represents the percentage of the number of patches whose density is less than the current density.

