# OpenReview forum: "Efficient Masked AutoEncoder for Video Object Counting and A Large-Scale Benchmark"
_ICLR.cc/2025/Conference — ICLR 2025 Poster_

### Official Review · Reviewer_rdoG · 2024-11-01

**Soundness:** 4
**Presentation:** 4
**Contribution:** 4
**Rating:** 8
**Confidence:** 5

**Summary:**

1. This paper addresses video object counting by introducing the DroneBird dataset for bird-counting scenarios and proposes a masked autoencoder-based counting framework.

2. The framework uses optical flow for temporal alignment and density maps for spatial focus to improve counting performance.

3. While each component is technically feasible, they lack integration to showcase a comprehensive advantage for crowd counting as a whole.

4. The dataset offers potential application value, though its academic innovation and contribution remain to be further substantiated.

**Strengths:**

1. Introduces a novel bird-counting dataset, DroneBird, captured from a drone’s perspective.

2. Proposes optical flow-guided Temporal Collaborative Fusion (TCF) to extract temporal information from adjacent frames, enhancing object counting accuracy.

3. Integrates Density-Embedded Masked Modeling (DEMO), using density maps as an auxiliary modality with image data, helping balance foreground and background in the model.

4. Employs Spatial Adaptive Masking (SAM) to focus on foreground areas, potentially improving model efficiency and counting accuracy.

**Weaknesses:**

1. The methodology's components (such as DEMO, SAM, and TCF) lack cohesive integration, appearing independently implemented without clearly demonstrated necessity for crowd counting.

2. Although DroneBird introduces a new category, the dataset’s difficulty and uniqueness are not strongly evident, and it lacks clear differentiation in research value compared to existing datasets beyond adding a bird-counting scenario.

**Questions:**

**Questions:**

1. In Table 1, video-based methods underperform image-based methods on the DroneBird dataset. What might be the reasons for this discrepancy, and could this be further elaborated?

2. Optical flow is typically more effective for capturing dynamic objects in scenes with a static camera. In the drone viewpoint of DroneBird, how does the model address the challenge where all objects in the field of view are in motion? Particularly, TCF’s improvements are minor according to the ablation study—how might this be explained?

3. How is Exp. III in Table 2 implemented? Given my understanding, SAM’s application without DEMO may lack practical relevance—how does it contribute in the absence of DEMO?

4. Could you further articulate the scientific value of DroneBird, especially in terms of the new challenges it introduces for video-based research? Specifically, what novel research questions or difficulties does it bring?

---

> ### Author Response · Authors · 2024-11-22
> **Response to Reviewer rdoG (1/6)**
>
> We sincerely thank the reviewer for your valuable comments and appreciate your recognizing on our **novel bird counting dataset**, **accuracy-enhanced**, **efficiency**, and **feasible model**. We provide detailed responses as follows:
>
> **Weakness 1: The methodology's components (such as DEMO, SAM, and TCF) lack cohesive integration, appearing independently implemented without clearly demonstrated necessity for crowd counting.**
>
> Thanks for the valuable comments. The proposed E-MAC is the first framework, which introduces the pre-trained vision foundation model to video object counting. We provide more explanations for the key components.
> 1. **Aligning with foundation model (DEMO)**: The proposed *DEMO* first takes the density map as an auxiliary modality to guide the token reconstruction, aligning the video object counting model with the multi-modal structure of Multi-MAE.
> 2. **Improve the masking strategy of DEMO by SAM**: The proposed **SAM is a key component of DEMO in our framework**. We statistically analyze the sparsity in the video data and present the results in Fig. 12 of Sec. A.7. We found that there is typical sparsity in the existing video data, which leads to a significant imbalance of the fore-background information. Therefore, the masking strategy of DEMO that balanced selects the fore- and back-ground is inefficient. Instead, we improve DEMO with SAM to adaptively perform masking according to the density map, assigning high mask probability for the sparse foreground regions, which effectively improves the counting performance.
> 3. **Multi-frame Fusion (TCF) for DEMO**: TCF is proposed to avoid potential uncertainty in some frames and improve the overall counting performance. It extends DEMO to cope with video counting tasks, using optical flow to temporally correlate neighboring frames and fusing the front and back frame information by calculating the residuals of the neighboring frames.

---

> ### Author Response · Authors · 2024-11-22
> **Response to Reviewer rdoG (2/6)**
>
> **Weakness 2: Although DroneBird introduces a new category, the dataset's difficulty and uniqueness are not strongly evident, and it lacks clear differentiation in research value compared to existing datasets beyond adding a bird-counting scenario.**
>
> Thanks for the valuable comment. Please kindly note that our DroneBird dataset is the first dynamic video counting dataset for small targets. **The dynamic flying bird is significant smaller than traditional crowd counting objects.** We calculated the number of pixels occupied by individual birds in the DroneBird dataset and the number of pixels occupied by individual humans in the existing crowd data. The number of pixels occupied by individual birds in DroneBird is significantly smaller than that occupied by individual humans in the crowd data (20 *vs.* 160), with the former being one-eighth of the latter.
>
> In addition, the proposed dataset does not just add a new counting category. Compared to video crowd counting, **birds move faster and tend to be more sparsely distributed in space, and these characteristics pose greater challenges.** In fact, the performance of the present method on bird counting scenarios is much weaker than its performance on crowd counting datasets. As shown in Table 1 in the main paper, the MAE and RMSE of the existing method on crowd counting are only in single digits, whereas they are much higher than those on the bird counting task, which is a significant gap. For example, the MAE and RMSE of the Gramformer [3] on the FDST dataset are 5.15 and 6.32, respectively, while the MAE and RMSE on the DroneBird dataset are 49.11 and 65.50. Moreover, this further validates the advantages of our proposed method over the existing methods: our method is ahead of the existing methods in both crowd and bird counting tasks.
>
> In addition, DroneBird indeed offers a new counting category. **One of the significant contributions of our new dataset is providing new target categories for the counting domain.** We observed that methods trained on other categories (mostly crowds) without fine-tuning or training directly failed to apply to bird counting tasks. As shown in the Table below, E-MAC trained on VSCrowd (Exp. 1) failed to perform well on the DroneBird testing set, while E-MAC trained on the DroneBird (Exp. 2) training set achieved a considerable performance. Therefore, an open bird counting dataset available for model training can provide additional data support for counting task research. The issue of bird conservation has been mentioned for a long time [1,2], and the proposal of DroneBird helps to assist in the analysis of bird activity, which in turn helps to conserve birds.
>
> |Exp.| Original dataset  | Target dataset  | MAE $\downarrow$ | RMSE $\downarrow$ |
> |:------:|:------:|:------:|:------:|:------:|
> |1|  VSCrowd  | DroneBird |  183.31  | 217.42 |
> |2|  DroneBird  | DroneBird |  38.72  | 42.92 |

---

> ### Author Response · Authors · 2024-11-22
> **Response to Reviewer rdoG (3/6)**
>
> **Question 1: In Table 1, video-based methods underperform image-based methods on the DroneBird dataset. What might be the reasons for this discrepancy, and could this be further elaborated?**
>
> Thanks for the valuable question. **We believe this may be due to the small proportion of birds in the images.**
>
> The data in DroneBird is captured from a drone's perspective, and to ensure that the birds' activities are not disturbed by the drone, the drone typically maintains a considerable distance from the bird flock, resulting in a small pixel proportion for individual birds.
> 1. Existing video methods focus more on how to leverage temporal information to improve model performance, not specifically designed for small-scale targets.
> 2. Single-frame methods often have special designs for small targets. For example, Gramformer [3] proposed a graph-modulated attention mechanism, which allows the model to focus on complementary information and effectively predict densities in smaller crowds, thus performing relatively better in counting small-scale targets.
> 3. Our method, considering the foreground-background imbalance issue, focuses more on foreground targets, thus copes to focus on a smaller proportion of prospect information and achieve superior performance.

---

> ### Author Response · Authors · 2024-11-22
> **Response to Reviewer rdoG (4/6)**
>
> **Question 2: Optical flow is typically more effective for capturing dynamic objects in scenes with a static camera. In the drone viewpoint of DroneBird, how does the model address the challenge where all objects in the field of view are in motion? Particularly, TCF's improvements are minor according to the ablation study—how might this be explained?**
>
> Thanks for the valuable question.
> 1. Depending on the actual requirements of the counting task and avoiding any impact on the bird's activities while capturing, **our data requires the drones to remain stationary or move only slowly during capturing**, so that there is no large motion impact for the camera.
> 2. In the ablated experiments, TCF is performed based on the baseline [4]. Since the baseline is simply performed on the pre-trained Multi-MAE weights, the single-frame estimation performance is limited. Thus, the multi-frame integration of TCF is hard to achieve a significant improvement. To fully validate the effect of TCF, we have added a new experiment by removing the TCF from the complete E-MAC (Exp. V in Table 2 of Sec. 5.3). In this experiment, the single-frame performance is promising. As shown in Table below, TCF improves the performance by 15%, which verifies the effectiveness of TCF.
>
> | Model | MAE $\downarrow$ | RMSE $\downarrow$ |
> |:------:|:------:|:------:|
> |  E-MAC w/o TCF  |   1.51  | 2.03 |
> |  E-MAC w/ TCF (Exp. V in Sec. 5.3) |   1.29  | 1.69 |

---

> ### Author Response · Authors · 2024-11-22
> **Response to Reviewer rdoG (5/6)**
>
> **Question 3: How is Exp. III in Table 2 implemented? Given my understanding, SAM's application without DEMO may lack practical relevance—how does it contribute in the absence of DEMO?**
>
> Thanks for the valuable question. Please kindly note that DEMO indicates the density-embedded masked modeling, which introduces the density map as an auxiliary modality to guide the masked patches reconstruction. SAM improves the masked modeling strategy by focusing on the foreground regions to overcome the sparse distribution of targets. **Without DEMO, the model degrades to the ViT framework instead of Multi-MAE framework**. Although it does not explicitly use density map as an auxiliary modality to instruct the patch reconstruction, we can also use the density map to perform adaptive and asymmetric sampling for the sparse fore- and back-ground.
>
> Therefore, the masking guidance of the image provided by the density map in our SAM is an independent process and does not rely on the density map as an auxiliary modal input to the encoder (DEMO). After removing the DEMO, the token with only the image modality is fed into the encoder, and the token with the image modality is still masked according to its corresponding density map (SAM). Briefly, in Exp.III, the input of the E-MAC *w/o* DEMO is a partially masked image. The input of the E-MAC *w/* DEMO is a partially masked image with a partially masked density map.

---

> > ### Comment · Reviewer_rdoG · 2024-11-22
> >
> > Thank you for your reply. I think that SAM only guides during the training phase within the original architecture. So, in the case of w/o DEMO, does it participate in the mask prediction during inference? After degrading to VIT, should the overall training mode be more like traditional supervised training rather than the mask autoencoder mode?

---

> > > ### Author Response · Authors · 2024-11-22
> > > **Further clarifications by Authors**
> > >
> > > Thank you for your quick reply! We provide further clarifications as follows:
> > > 1. Yes, the adaptive masking of SAM only provides guidance during the training phase, and it does not participate in predition during inference. Please kindly note that, during inference, we take all the tokens from image modality as input without any density maps, which means the tokens from density map modality are totally masked.
> > > 2. After degrading to ViT, the overall training mode is a masked cross-modal prediction. The difference between it and mask autoencoder is that the modalities of input and output are different. Meanwhile, different from traditional supervised training, the model is trained as masked token regression.
> > >
> > > Thanks again for your constructive comments!

---

> > > > ### Comment · Reviewer_rdoG · 2024-11-22
> > > >
> > > > Thanks for the detailed response from the authors. This helps to solve my concerns, so I have increased my rating accordingly.

---

> > > > > ### Author Response · Authors · 2024-11-22
> > > > > **Thank you!**
> > > > >
> > > > > Thanks for your reply! We are delighted to have addressed your concerns and appreciate your insightful comments and open discussions, which have greatly improved our work and inspired us to research more. If you have any further questions or suggestions, please feel free to post, and we will continually work on this project and actively address your concerns.

---

> ### Author Response · Authors · 2024-11-22
> **Response to Reviewer rdoG (6/6)**
>
> **Question 4: Could you further articulate the scientific value of DroneBird, especially in terms of the new challenges it introduces for video-based research? Specifically, what novel research questions or difficulties does it bring?**
>
> Thank for the valuable question. We summarize the contributes and new challenges brought by DroneBird into the following four points:
> 1. **Small target**：We compared the number of foreground pixels occupied by independent targets in the DroneBird data and the existing crowd data at the same resolution, and the independent targets in DroneBird occupy fewer pixels (**less than 20 pixels hight**) than the independent targets in the existing dataset (**more than 160 pixels hight**). Note that although the counting target in the crowd counting task is the human head, the human body still provides important information and the human body occupies more pixels compared to birds, so our dataset provides data on small targets for counting. On the other hand, in order to avoid drone interference with bird activity, DroneBird's drone was photographed farther away from the flock, resulting in a further reduction of pixels occupied by the photographed individual birds.
> 2. **Sparse distribution**：Unlike humans, birds in nature need enough space to move around, which means birds are more sparsely distributed in space. Coupled with the smaller target size of birds, the sparseness of the foreground portion is even more pronounced in DroneBird compared to the already existing dataset.
> 3. **Fast movement**: Unlike existing video crowd-sourced data, the unique flight movements of birds allow them to do fast movements, posing a challenge for better temporal modeling.
> 4. **New counting category**:
>
>     **DroneBird fills the gap of missing video bird data in the field.** In fact, the performance of the existing method on the bird counting dataset is much weaker than its performance on the crowd counting dataset. As shown in Table 1 in the main paper, the MAE and RMSE scores of most recent methods are commonly in single digits on crowd counting dataset, while they are mainly much larger than 30 in the bird counting task. For example, the MAE and RMSE of the Gramformer [3] on the FDST dataset are 5.15 and 6.32, respectively, while the MAE and RMSE on the DroneBird dataset are 49.11 and 65.50, which is a significant gap between the two counting tasks. It further verified the advantages of our proposed method over the existing methods. Our method achieves considerable performance on video crowd datasets while also leading existing methods on the more challenging DroneBird dataset.
>
>     In addition, the models trained on crowd data fail to conduct bird counting directly (See Table below). We evaluated the performance of our method trained on the VSCrowd dataset (Exp. 1) and the DroneBird training set (Exp. 2) on the DroneBird testing set. The results show that the model trained on the VSCrowd dataset failed to perform well on the bird data, which indicates the large gap between crowd and bird data.
>
> We believe that DroneBird is qualified to alleviate the current scarcity of bird data in the field, supporting further bird conservation activities.
>
> |Exp.| Original dataset  | Target dataset  | MAE $\downarrow$ | RMSE $\downarrow$ |
> |:------:|:------:|:------:|:------:|:------:|
> |1|  VSCrowd  | DroneBird |  183.31  | 217.42 |
> |2|  DroneBird  | DroneBird |  38.72  | 42.92 |

---

> ### Author Response · Authors · 2024-11-22
>
> # References
>
> [1] Bairlein, Franz. "Migratory birds under threat." Science 354.6312 (2016): 547-548.
>
> [2] Lees, Alexander C., et al. "State of the world's birds." Annual Review of Environment and Resources 47.1 (2022): 231-260.
>
> [3] Lin, Hui, et al. "Gramformer: Learning Crowd Counting via Graph-Modulated Transformer." Proceedings of the AAAI Conference on Artificial Intelligence. Vol. 38. No. 4. 2024.
>
> [4] Bachmann, Roman, et al. "Multimae: Multi-modal multi-task masked autoencoders." European Conference on Computer Vision. Cham: Springer Nature Switzerland, 2022.

---

### Official Review · Reviewer_Dnim · 2024-11-03

**Soundness:** 3
**Presentation:** 2
**Contribution:** 3
**Rating:** 6
**Confidence:** 5

**Summary:**

This paper presents the E-MAC framework, which addresses the dynamic imbalance of fore-background in video object counting through density-embedded Efficient Masked Autoencoder Counting. By leveraging optical flow-based temporal collaborative fusion, E-MAC aligns features across frames to improve counting accuracy. The approach introduces Density-Embedded Masked Modeling (DEMO) to enhance intraframe representation of dynamic foreground objects. To mitigate redundant background information, an efficient spatial adaptive masking derived from density maps is employed. Additionally, the paper introduces DroneBird, a large-scale bird counting dataset for migratory bird protection, showcasing the framework’s effectiveness in natural scenarios and validating its superiority through extensive experiments on three crowd datasets.

**Strengths:**

1.The efficient spatial adaptive masking method adeptly handles dynamic density distribution, ensuring focus on foreground regions and tackling the imbalance of fore-background effectively.

2.Introduction of the DroneBird dataset, the first video bird counting dataset, significantly broadens the application domain, demonstrating the framework’s versatility and impact through extensive validation against competing methods.

**Weaknesses:**

1.The manuscript lacks visual analyses of the three key components, which would illustrate the targeted effectiveness of each module.

2.It is recommended to provide code or critical sections of the code for review to enable thorough evaluation of the methodology and results.

**Questions:**

1.In Table 1, do the compared methods use the same backbone network (ViT-B) as this paper's method? If not, please specify the reasons.

2.Why does the proposed method show higher RMSE on the VSCrowd dataset compared to GNANet in Table 1?

---

> ### Author Response · Authors · 2024-11-22
> **Response to Reviewer Dnim (1/4)**
>
> We'd like to thank the reviewer for the valuable comments, and acknowledgment of our method **tackling the imbalance of fore-background effectively**, **versatility**, and **our dataset significantly broadens the application domain**. We believe the constructive feedback will improve the paper and increase its potential impact on the community.
>
> **Weakness 1: The manuscript lacks visual analyses of the three key components, which would illustrate the targeted effectiveness of each module.**
>
> Thanks for the valuable comment. **We visualized some of the predicted results of our key components (TCF, DEMO, SAM) and our baseline [1], as shown in Fig. 11 in Line 978 of the Appendix.** We sequentially added our key components (TCF, DEMO, SAM) to the baseline, and observed that the visualization effects were continuously improving.
>
> Additionally, we have also included some visual effects of the TCF module in Fig. 9 of Sec.A.5 in the Appendix. Additionally, we provided the difference of predicted density map *w/* and *w/o* using SAM and the comparison of the number of foreground and background image patches entering the encoder *w/* and *w/o* using SAM in Fig. 10 of Sec.A.6 in Appendix. We hope these visual analysis can help reviewers to better understand the effectiveness of each module.
> 1) In Fig. 9, we show the visual effects of the TCF module. Obviously, the TCF module effectively eliminates some key background interference.
> 2) In Fig. 10, we compare the prediction difference of the same image when trained with different masking strategies. The results show that the model trained with SAM has a more accurate prediction of the targets, while the model trained w/o SAM failed to predict some targets.
> 3) In Fig. 10, we also compare the number of foreground and background image patches entering the encoder *w/* and *w/o* using SAM. It is evident that *w/* SAM, there was a significant imbalance in the background image patches entering the encoder. Using only the foreground part resulted in slightly more foreground image patches than the background entering the encoder. Therefore, we introduced the background retention probability (BRP) to adjust the proportion of foreground image patches feeding to the encoder, ensuring a more balanced input of foreground and background image patches for model learning.

---

> ### Author Response · Authors · 2024-11-22
> **Response to Reviewer Dnim (2/4)**
>
> **Weakness 2: It is recommended to provide code or critical sections of the code for review to enable thorough evaluation of the methodology and results.**
>
> Thanks for the valuable comment. All the code will be definitely released, and in fact our code is already ready. **We have provided the code for the proposed method in the anonymous [URL](https://anonymous.4open.science/r/E-MAC-4631/README.md) for review**. The code includes the implementation details of the proposed method, pretrained weights and the evaluation process. We hope this will help reviewers better understand the methodology and results.

---

> ### Author Response · Authors · 2024-11-22
> **Response to Reviewer Dnim (3/4)**
>
> **Question 1: In Table 1, do the compared methods use the same backbone network (ViT-B) as this paper's method? If not, please specify the reasons.**
>
> Thanks for the constrcutive question. Please kindly note that **we are the first work to introduce the pre-trained vision foundation model to video object counting task**. Although few of existing methods use the same backbone as ours, we have added experiments on introducing the pre-trained model to some existing methods, such as STGN [7] and Gramformer [6]. As shown in the Tabel below, STGN and Gramformer using the pre-trained MultiMAE weights only achieve 4.04, 4.19 (Exp. 1) and 5.15, 6.32 (Exp. 2) on MAE and RMSE, respectively. While our method achieves 1.29 and 1.69 of MAE and RMSE (Exp. 3), which is a significant improvement. In Exp. 5, although using the pre-trained vanilla ViT weights, our method still achieves 1.49 and 1.93 of MAE and RMSE.
>
> |Exp.|  Model | MAE $\downarrow$  | RMSE $\downarrow$ |
> |:------:|:------:|:------:|:------:|
> |1| STGN w/ MultiMAE | 4.04| 4.19 |
> |2| Gramformer w/ MultiMAE | 5.15 | 6.32 |
> |3|  E-MAC w/ MultiMAE (Exp.V in Sec.5.3)  |  1.29  | 1.69 |
> |4| vanilla ViT-B | 2.63 | 3.31 |
> |5|  E-MAC w/ vanilla ViT-B | 1.49 | 1.93 |
>
> Specifically, most of existing methods can be typically classified into three categories: extracting multi-scale features by improving the structural designs [2,3,4], focusing on the scale-variation problem by introducing attention-based methods [5,6], and temporal information ensemble methods [7,8,9]. These works focuses on addressing scale variations as well as temporal issues. Unlike these methods, we for the first time explore the impact of the founditional model on the counting task and present the first framework based on the pre-trained vision foundation model. We use the density map as an auxiliary modality for efficient sampling in the form of asymmetric sampling to achieve efficient learning of image-to-density map regression. We conduct a series of experiments (Sec. 5) to demonstrate the effectiveness of our method. Please kindly note that although the pretrained ViT model provides comparable performance, our E-MAC even further improve more than 40% (Exp. 4 *vs.* Exp. 5). **Our approach extends the great potential of the visual foudation model for counting tasks and provides a new perspective for the community.**

---

> ### Author Response · Authors · 2024-11-22
> **Response to Reviewer Dnim (4/4)**
>
> **Question 2: Why does the proposed method show higher RMSE on the VSCrowd dataset compared to GNANet in Table 1?**
>
> Thanks for the valuable question. Please kindly note that **GNANet [10] adopts more frames than ours during prediction.** Specifically, GNANet uses both the previous, current and next frames for temporal modeling (offline), while our method only uses the previous frame and the current frame (online). As a comparison, our method still maintains competitive performance in RMSE and performs better on most images with lower MAE compared to GNANet (27% improvement).

---

> ### Author Response · Authors · 2024-11-22
>
> # References
>
> [1] Bachmann, Roman, et al. "Multimae: Multi-modal multi-task masked autoencoders." European Conference on Computer Vision. Cham: Springer Nature Switzerland, 2022.
>
> [2] Zhang, Yingying, et al. "Single-image crowd counting via multi-column convolutional neural network." Proceedings of the IEEE conference on computer vision and pattern recognition. 2016.
>
> [3] Li, Yuhong, Xiaofan Zhang, and Deming Chen. "Csrnet: Dilated convolutional neural networks for understanding the highly congested scenes." Proceedings of the IEEE conference on computer vision and pattern recognition. 2018.
>
> [4] Liu, Weizhe, Mathieu Salzmann, and Pascal Fua. "Context-aware crowd counting." Proceedings of the IEEE/CVF conference on computer vision and pattern recognition. 2019.
>
> [5] Lin, Hui, et al. "Boosting crowd counting via multifaceted attention." Proceedings of the IEEE/CVF conference on computer vision and pattern recognition. 2022.
>
> [6] Lin, Hui, et al. "Gramformer: Learning Crowd Counting via Graph-Modulated Transformer." Proceedings of the AAAI Conference on Artificial Intelligence. Vol. 38. No. 4. 2024.
>
> [7] Wu, Zhe, et al. "Spatial-temporal graph network for video crowd counting." IEEE Transactions on Circuits and Systems for Video Technology 33.1 (2022): 228-241.
>
> [8] Hou, Yi, et al. "Frame-recurrent video crowd counting." IEEE Transactions on Circuits and Systems for Video Technology 33.9 (2023): 5186-5199.
>
> [9] Fang, Yanyan, et al. "Locality-constrained spatial transformer network for video crowd counting." 2019 IEEE international conference on multimedia and expo (ICME). IEEE, 2019.
>
> [10] Li, Haopeng, et al. "Video crowd localization with multifocus gaussian neighborhood attention and a large-scale benchmark." IEEE Transactions on Image Processing 31 (2022): 6032-6047.

---

### Official Review · Reviewer_P9ZV · 2024-11-04

**Soundness:** 2
**Presentation:** 1
**Contribution:** 2
**Rating:** 5
**Confidence:** 2

**Summary:**

This paper presents a method and a new dataset for video object counting. The model performs well in multiple object-counting benchmarks using video as input.

**Strengths:**

- A video object counting benchmark, Dronebird, is established.
- A corresponding object counting method to deal with video data is proposed.

**Weaknesses:**

Many descriptions are not inconsistent, making it difficult to understand the paper. See Questions.

**Questions:**

- In video counting, how is the count of a video defined? Is it the average count of objects, or the number of different instances that appear in the video?
- The TCF module presented in Fig. 2 is inconsistent with the corresponding description. In Eq. (1), the product of the "warp" operator is added to $\hat{D}_t$ to generate $\hat{D}_{\text{fuse}}$, and the "warp" operation involves $\hat{D}_t$. In the description, a cross-attention is applied to generate $\hat{D}_t^{\text{res}}$, which is then combined with $\hat{D}_t$ to produce $\hat{D}_{\text{fuse}}$. In Fig. 2, the generation of $\hat{D}_t^{\text{res}}$ does not involve $\hat{D}_t$ according to the arrows that demonstrate the data flow.
- In Eq. (2), $\mathbf{T}_D \in \mathbb{B}^{B \times \mathcal{N}_D \times C}$ is a token set including vectors. How are these vectors sorted?
- The description of SAM is also different from that in Fig. 2. The description takes the ground truth $D_t$ to mask tokens, while SAM in Fig. 2 involves $I_{t-1}$, $D_{t-1}$, and a random mask, which does not appear in Sec. 4.3.
- During training, $\mathbf{S}_t = \{I_t, D_t\}$ contains the input image and GT labels. How does the model work in inference, without GT as input?

---

> ### Author Response · Authors · 2024-11-22
> **Response to Reviewer P9ZV (1/5)**
>
> We would like to thank the reviewer for the thoughtful and thorough comments on our paper as well as for recognizing our **new proposed counting dataset** and **well performance**. We will also make an effort to increase clarity throughout. Below, we provide the point-to-point responses:
>
> **Question 1: In video counting, how is the count of a video defined? Is it the average count of objects, or the number of different instances that appear in the video?**
>
> Thanks for the valuable question. According to the previous works [1,2], **the target of video counting is to predict the number of objects in each frame of the video.** We follow this setting and define the target of video counting as the number of objects in each frame. For the evaluation of counting results, video counting calculates the difference between the predicted results and the ground truth for each frame, and then computes the mean absolute error (MAE) and mean squared error (RMSE) across all frames.  For example, given a sequence of 100 video frames, the goal of video counting on this sequence of video frames is to predict the number of targets on each frame. For this sequence, the result of video counting will be an array of length 100, where each element represents the number of targets on the corresponding frame. Assuming the predicted result for each frame is $\hat{D}\_i$ and the corresponding ground truth is $D_i$, then the evaluation metric MAE (Mean Absolute Error) is calculated as $MAE = 1/100*\sum_{i=1}^{100} |\hat{D}\_i-D_i|$, RMSE (Root Mean Squared Error) is calculated as $RMSE = \sqrt{1/100*\sum_{i=1}^{100} (\hat{D}\_i-D_i)^2}$.

---

> ### Author Response · Authors · 2024-11-22
> **Response to Reviewer P9ZV (2/5)**
>
> **Question 2: The TCF module presented in Fig. 2 is inconsistent with the corresponding description. In Eq. (1), the product of the "warp" operator is added to $\hat{D}\_{t}$ to generate $\hat{D}\_{\text{fuse}}$, and the "warp" operation involves $%\hat{D}\_t$. In the description, a cross-attention is applied to generate $\hat{D}\_t^{\text{res}}$, which is then combined with $\hat{D}\_t$ to produce $\hat{D}\_{\text{fuse}}$. In Fig. 2, the generation of $\hat{D}\_t^{\text{res}}$ does not involve $\hat{D}\_t$ according to the arrows that demonstrate the data flow.**
>
> Thanks for the valuable question. In Figure 2 and Eq. 1, we use a ''*W* '' to simultaneously represent the warp and cross-attention operations. In Figure 2, $\hat{D}_t$, $\hat{D}\_{t-1}$, and optical flow are all connected to ''*W* '', indicating the use of optical flow, $\hat{D}\_t$, and $\hat{D}\_{t-1}$ to generate $\hat{D}\_{res}$, which is consistent with Eq. 1. We have also provided the complete training pseudo code in the appendix for detailed explanation. The step-by-step operations of the TCF module are summarized as follows:
> 1. $O = \phi\_{opticalflow}(I\_t, I\_{t-1})$
> 2. $\hat{D}\_{t-1}^{warp} = \phi\_{warp}(O, \hat{D}\_{t-1})$
> 3. $\hat{D}\_{res} = \phi\_{ca}(\hat{D}\_t, \hat{D}\_{t-1}^{warp})$
> 4. $\hat{D}\_{fuse} = \hat{D}\_{res} + \hat{D}\_t$
>
> We will emphasize this in the revision to avoid potential confusing. Thanks again for the constructive comment.

---

> ### Author Response · Authors · 2024-11-22
> **Response to Reviewer P9ZV (3/5)**
>
> **Question 3: in Eq.(2), $T_D \in B^{B\times N_D\times C}$ is a token set including vectors. How are these vectors sorted?**
>
> Thanks for the valuable question. **We have also provided a detailed Figure (Fig. 8) in Sec.A.2 of the Appendix.** Assuming we have got the number of tokens to be retained ($N_I^{ret}, N_D^{ret}$), we further explain our sorting operation on $T_D$ based on Fig. 8 as follows:
> 1. For an image-density map pair, the image and density map are divided into patches (for easier understanding, we divided the image and density map into 25 patches in Fig. 8).
> 2. Patches from the image and density map in the same position are paired. Patch pairs are then processed into token pairs ($T_I$, $T_D$).
> 3. For each patch pair, we use the integral value of the density map patch, which represents the number of objects in the image patch, to represent the weight of the corresponding token pair.
> 4. **We sort the token pairs based on the weight calculated in the step. 3** and obtain the original token indices before sorting.
> 5. For $T_I$, we retain the top $N_I^{ret}$ tokens in the sorted $T_I$.
> 6. For $T_D$, we shuffle the order of $T_D$ and retain the first $N_D^{ret}$ tokens.
> 7. We then concatenate the retained tokens from $T_I$ and $T_D$ and shuffle its order to form the final token set.

---

> ### Author Response · Authors · 2024-11-22
> **Response to Reviewer P9ZV (4/5)**
>
> **Question 4: The description of SAM is also different from that in Fig. 2. The description takes the ground truth $D_t$ to mask tokens, while SAM in Fig. 2 involves $I_{t-1}$, $D_{t-1}$, and a random mask, which does not appear in Sec. 4.3.**
>
> Thanks for the valuable question. For the density map modality, at any stage of training, we use a Dirichlet distribution to determine the number of tokens to retain and apply random masking to the density map, as we mentioned in Line 243 of Sec. 4.2. **To avoid confusing, we have provided more explanations in Sec. 4.3.** We use the density map to guide the foreground masking probability of the corresponding image modality, thereby achieving adaptive masking. For the time steps t and t-1 in Figure 2, since t-1 is located in the first layer, we use the variables from time step t-1 to describe the SAM process in the figure. In fact, for time step t, the subscript of the variables in the SAM module should be changed from t-1 to t. To avoid confusing, we have standardized all the descriptions in figure and the main text. Thanks again for the constructive comments.

---

> ### Author Response · Authors · 2024-11-22
> **Response to Reviewer P9ZV (5/5)**
>
> **Question 5: During training, $S_t=I_t,D_t$ contains the input image and GT labels. How does the model work in inference, without GT as input?**
>
> Thanks for the question. Since the training process requires masking operations on both images and density maps, the masking rates for images and density maps are obtained by a Dirichlet distribution. It is conceivable that in some cases, all image tokens are retained while all density map tokens are discarded. When all image tokens are retained, there is no need for GT to perform dynamic masking for images, which is consistent with the data input during the inference phase.
>
> Please kindly note that, during the inference phase, **we only input the images from the previous and current frames, indicating all the density map patches are masked**. For the decoder, we use random tokens to complete masked tokens in $T_D$. We will emphasize this in the revision to avoid confusing.

---

> ### Author Response · Authors · 2024-11-22
>
> # References
> [1] Wu, Zhe, et al. "Spatial-temporal graph network for video crowd counting." IEEE Transactions on Circuits and Systems for Video Technology 33.1 (2022): 228-241.
>
> [2] Hou, Yi, et al. "Frame-recurrent video crowd counting." IEEE Transactions on Circuits and Systems for Video Technology 33.9 (2023): 5186-5199.

---

> ### Comment · Reviewer_P9ZV · 2024-11-23
> **Further Comments**
>
> **RQ1.** The explanation is clear but it would be better to describe it in the main paper.
>
> **RQ2.** It is confusing to use $\phi_{(\cdots)}$ to denote the operation in the pseudo code, but a "W in a circle" to represent these operations in Fig. 2. It would be better to replace "W" with $\phi$ in Fig. 2 or replace $\phi$ with $W$ in the pseudo code.
>
> **RQ3.** It is clear in the response but not clear in the paper:
> 1. I noticed that you do not mention these details in the revised paper (description about Fig. 8). According to the added content (from line 856 to line 859), I still cannot understand it. You'd better add these contents to the paper since it is really confusing if there is no detailed description.
> 2. Accordingly, BRP in line 269 is actually the count in each patch? No matter right or wrong, it would be better to describe it in the paper. BRP is confusing but the response here is clear.
> 3. Accordingly, Eq.(2) should be formulated according to your description. Eq.(2) does not provide any information about how to compute BRP or the count in the patch.
> 4.  What is the work of the shuffle operation in Fig. 8? For a transformer-based decoder, I think the precedence of tokens does not affect the computation formulation and each token should also be embedded with positional information.
>
> **RQ4.** No, you misunderstand my meaning. In Sec. 4.1, you use $D$ to denote GT (line 212), and use $\hat{D}$ to represent the predicted density map (line 221). In line 265, you use $D$ here, so I consider you are using GT as input.
>
> **RQ5.** Accordingly, "For the decoder, we use random tokens to complete masked tokens in $T_D$."
> 1. How is the mask implemented? In line 15 of Algorithm 2?
> 2. Will the performance be affected if different random seeds are used?

---

> > ### Author Response · Authors · 2024-11-25
> > **Response to Further Comments (1/8)**
> >
> > **RQ1. The explanation is clear but it would be better to describe it in the main paper.**
> >
> > Thanks for your invaluable comments. We have updated the detailed explanation in the main paper, as shown in Line 128-131 of the main paper.

---

> > ### Author Response · Authors · 2024-11-25
> > **Response to Further Comments (2/8)**
> >
> > **RQ2. It is confusing to use $\phi_{(\cdots)}$ to denote the operation in the pseudo code, but a "W in a circle" to represent these operations in Fig. 2. It would be better to replace "W" with $\phi$ in Fig. 2 or replace $\phi$ with $W$ in the pseudo code.**
> >
> > Thanks for your constructive suggestion. We have replaced the "W in a circle" in the original Figure 2 by $\phi_{warp}$ and $\phi_{ca}$, and updated Figure 2 in the main paper.

---

> > ### Author Response · Authors · 2024-11-25
> > **Response to Further Comments (3/8)**
> >
> > **RQ3. It is clear in the response but not clear in the paper:**
> >
> > **1. I noticed that you do not mention these details in the revised paper (description about Fig. 8). According to the added content (from line 856 to line 859), I still cannot understand it. You'd better add these contents to the paper since it is really confusing if there is no detailed description.**
> >
> > Thanks for your invaluable comments. We have provided further explanations in the revised paper as follows:
> >
> > For ease of expression, we crop the image and the density map into 25 patches, as shown in Fig. 8, this number varies according to the size of input images (each path is set to $16\times16$). We first calculate the $\mathbf{V}\_D^i$ of each density map patch $\mathbf{D}\_{patch}^i$, and sort the token according to the number of targets $\mathbf{V}\_D^i$. To balance the fore- and back-ground information, we set a background retention probability (BRP) $\mathcal{P}$ to determine the sorting manner, which is detailed in Sec. 4.3. For tokens from image modality, we keep the first $\mathcal{N}\_I^\text{ret}$ tokens after sorting. For tokens from density map modality, we randomly shuffle their order and keep the first $\mathcal{N}\_I^\text{ret}$ tokens, i.e., randomly select $\mathcal{N}\_I^\text{ret}$ density tokens.
> > Since the masked tokens are filled by learnable tokens in the decoder according to the pre-trained foundation model [3], we first restore the retained image and density tokens to their original order before sorting. Then, we concatenate and feed them into the encoder. Note that, the restoration can be performed in the decoder as well.
> > The full set of retained tokens and filled density map tokens are then fed into the decoder to predict the density map. Specifically, the full set of retained tokens are treated as $key$ and $value$ vectors, and the filled density map tokens are treated as $query$ vector in the cross-attention layer in the decoder. Then, the output of the cross-attention layer is fed into two self-attention layers to perform the final prediction.
> >
> > We have updated the description of Fig. 8 in Sec. A.2 of the Appendix.

---

> > ### Author Response · Authors · 2024-11-25
> > **Response to Further Comments (4/8)**
> >
> > **RQ3. 2. Accordingly, BRP in line 269 is actually the count in each patch? No matter right or wrong, it would be better to describe it in the paper. BRP is confusing but the response here is clear.**
> >
> > Thanks for your valuable questions and constructive suggestions. The count in the $i$-th patch is represented by $\mathbf{V}^i_D$. We calculate the number of targets $\mathbf{V}^i\_D$ in the $i$-th density map patch $\mathbf{D}^i\_\text{patch}$ corresponding to each token, and $\mathbf{V}\_D = ${$\mathbf{V}\_D^1,\mathbf{V}\_D^2,\cdots, \mathbf{V}\_D^{\mathcal{N}\_I}$}, where $\mathbf{V}\_D^i = \phi\_{\text{sum}}(\mathbf{D}\_\text{patch}^i)$. $\phi\_{\text{sum}}$ represents the pixel-wise sum operation in each density patch [1,2], and the results represent the number of targets in the corresponding patch. To focus on the foreground, we sort the image tokens according to the number of targets $\mathbf{V}^i\_D$ in the corresponding density map modality. While the foreground provides more valid information, the background should not be completely ignored. Therefore, **we set a **b**ackground **r**etention **p**robability (**BRP**) $\mathcal{P}$ to introduce the background information, where BRP determines the sorting manner, in ascending order with a probability of $\mathcal{P}$ (focus on the background) or in descending order with a probability of $1$ - $\mathcal{P}$ (focus on the foreground)**.
> >
> > We have updated the description in Line 270-280 of the main paper, and conducted ablated experiments in Line 499-509 to explore the threshold for BRP in our experiments.

---

> > ### Author Response · Authors · 2024-11-25
> > **Response to Further Comments (5/8)**
> >
> > **RQ3.3. Accordingly, Eq.(2) should be formulated according to your description. Eq.(2) does not provide any information about how to compute BRP or the count in the patch.**
> >
> > Thanks for the valuable comments. We have updated Eq. (2) and provided more details for computing the count in the patch.
> >
> > Specifically, the updated Eq. (2) is as follow:
> > $$
> > \\mathcal{K} = \\left\\{
> >    \\begin{aligned}
> >     \\texttt{argsort}\_{des}(\\phi\_{\\text{sum}}(\\mathbf{D}\_\\text{patch}^i))\\{1:\\mathcal{N}\_I^\\text{ret}\\},& \\quad \\text{if} \\quad \\mathbb{N} \\leq 1-\\mathcal{P},\\\\
> >     \\texttt{argsort}\_{asc}(\\phi\_{\\text{sum}}(\\mathbf{D}\_\\text{patch}^i))\\{1:\\mathcal{N}\_I^\\text{ret}\\},& \\quad \\text{otherwise},
> >     \\end{aligned}
> > \\right.
> > $$
> >
> > Specifically, we denote the density map patch corresponding to the $i$-th token in $\mathbf{T}\_D$ as $\mathbf{D}\_\text{patch}^i$. **We use a pixel-wise summation operation $\phi\_{\text{sum}}$ to calculate the sum of the density map patch $\mathbf{D}\_\text{patch}^i$ as $\phi\_{\text{sum}}(\mathbf{D}\_\text{patch}^i)$, which represents the count of targets in the corresponding image patch.** Then we perform the sorting operation $\texttt{argsort}$ to get the indices of tokens after sorting, where we perform ascending sorting if the random number is less than $1-\mathcal{P}$, and descending sorting otherwise, where $\mathcal{P}$ is the BRP as a hyperparameter. We then retain tokens according to the first $\mathcal{N}\_I^\text{ret}$ indices.

---

> > ### Author Response · Authors · 2024-11-25
> > **Response to Further Comments (6/8)**
> >
> > **RQ3.4. What is the work of the shuffle operation in Fig. 8? For a transformer-based decoder, I think the precedence of tokens does not affect the computation formulation and each token should also be embedded with positional information.**
> >
> > Thanks for your detailed question. We have provided more details of the shuffle operation in Fig. 8.
> > Specifically, to balance the fore- and back-ground information, we set a background retention probability (BRP) $\mathcal{P}$ to determine the sorting manner, which is detailed in Sec. 4.3. For tokens from image modality, we keep the first $\mathcal{N}_I^\text{ret}$ tokens after sorting. For tokens from density map modality, we randomly **shuffle** their order and keep the first $\mathcal{N}_I^\text{ret}$ tokens, i.e., randomly select $\mathcal{N}_I^\text{ret}$ density tokens. Here, to aviod confusing, we replace the left "*Shuffle*" in the original figure to "*Restore*". Since the masked tokens are filled by learnable tokens in the decoder according to the pre-trained foundation model [3], we first restore the retained image and density tokens to their original order before sorting. Then, we concatenate and feed them into the encoder.
> >
> > It is worth noting that the restoration can be performed in the decoder as well. As we add positional encoding to the tokens, the precedence of tokens does not affect the computation, but is simply to enable easier implementation of filling masked tokens.
> > The detailed revisions have been updated in L270-293 of the main paper and L836-851, L885-898 of the supplementary material.

---

> > ### Author Response · Authors · 2024-11-25
> > **Response to Further Comments (7/8)**
> >
> > **RQ4. No, you misunderstand my meaning. In Sec. 4.1, you use $D$ to denote GT (line 212), and use $\hat{D}$ to represent the predicted density map (line 221). In line 265, you use $D$ here, so I consider you are using GT as input.**
> >
> > Thanks for your question. Our SAM dynamically samples the image according to the foreground of density map, such that, we adopt $D$ (GT) as an indicator of the foreground during training. During inference, the SAM will be removed and there is no GT will be fed into the network. In addition, **we have provided detailed computation description of spatial adaptive mask in L285-298 of the main paper**.
> >
> > Specifically, to obtain the spatial adaptive mask $M\_\text{adaptive}=\\{M\_\text{adaptive}^i | 1 \leq i \leq \mathcal{N}\_I\\}$  for image $I$. For each token in position $i$ and its corresponding mask $M^{i}\_\text{adaptive}$, we have
> > $$
> >     M^{i}\_\text{adaptive} = \\left\\{
> >     \\begin{aligned}
> >         0, & \\quad  \\text{if} \\quad i \\in \\mathcal{K},\\\\
> >         1, & \\quad  \\text{otherwise},
> >     \\end{aligned}
> >     \\right.
> > $$
> > where $0$ represent keeping and $1$ represent masking. We denote the retained tokens from the image $I$ as $\mathbf{T}\_I^\text{ret} \in \mathbb{R}^{B \times \mathcal{N}\_I^\text{ret} \times C}$, where $\mathbf{T}\_I^\text{ret} = \mathbf{T}\_I \otimes (1-M\_\text{adaptive})$ and $\mathbf{T}\_I$ denotes all the tokens of image $I$. For the density token $\mathbf{T}\_D$ corresponding to density map patch $\mathbf{D}\_\text{patch}$, we generate a *random mask* $M\_\text{random}$ to retain $\mathcal{N}\_D^\text{ret}$ tokens as $\mathbf{T}\_D^\text{ret} \in \mathbb{R}^{B \times \mathcal{N}\_D^\text{ret} \times C}$. These retained tokens are then concatenated to $\mathbf{T}^\text{ret} \in \mathbb{R}^{B \times (\mathcal{N}\_I^\text{ret}+\mathcal{N}\_D^\text{ret}) \times C}$ and fed into the decoder for prediction.
> >
> > During inference, the density maps tokens are fully masked and removed. The image tokens are fed into the model alone to reconstruct the density maps. In other words, we set $\mathcal{N}\_D^{\text{ret}} = 0$ and $\mathcal{N}\_I^{\text{ret}} = \mathcal{N}\_I$.

---

> > ### Author Response · Authors · 2024-11-25
> > **Response to Further Comments (8/8)**
> >
> > **RQ5. Accordingly, "For the decoder, we use random tokens to complete masked tokens in $T\_D$."**
> >
> > **1. How is the mask implemented? In line 15 of Algorithm 2?**
> >
> > Thanks for the detailed questions. **We have provided detailed computation description of mask in L285-293 of the main paper and updated Algorithm 2.** Please kindly note that the line 15 of original Algorithm 2 is the line 16 of the current version.
> >
> > Specifically, to obtain the spatial adaptive mask $M\_\text{adaptive}=\\{M\_\text{adaptive}^i | 1 \leq i \leq \mathcal{N}\_I\\}$  for image $I$. For each token in position $i$ and its corresponding mask $M^{i}\_\text{adaptive}$, we have
> > $$
> >     M^{i}\_\text{adaptive} = \\left\\{
> >     \\begin{aligned}
> >         0, & \\quad  \text{if} \\quad i \in \mathcal{K},\\\\
> >         1, & \\quad  \text{otherwise},
> >     \\end{aligned}
> >     \\right.
> > $$
> > where $0$ represent keeping and $1$ represent masking. We denote the retained tokens from the image $I$ as $\mathbf{T}\_I^\text{ret} \in \mathbb{R}^{B \times \mathcal{N}\_I^\text{ret} \times C}$, where $\mathbf{T}\_I^\text{ret} = \mathbf{T}\_I \otimes (1-M\_\text{adaptive})$ and $\mathbf{T}\_I$ denotes all the tokens of image $I$. For the density token $\mathbf{T}\_D$ corresponding to density map patch $\mathbf{D}\_\text{patch}$, we generate a *random mask* $M\_\text{random}$ to retain $\mathcal{N}\_D^\text{ret}$ tokens as $\mathbf{T}\_D^\text{ret} \in \mathbb{R}^{B \times \mathcal{N}\_D^\text{ret} \times C}$.
> >
> > In addition, we have also released the [code](https://anonymous.4open.science/r/E-MAC-4631/README.md) for ease of understanding of our approach. Specifically, our implementation code for the mask (line 16 of current Algorithm 2) is as follows:
> >
> > ```python
> > input_tokens = torch.gather(
> >     input_tokens,
> >     dim=1,
> >     index=ids_keep.unsqueeze(-1).repeat(1, 1, input_tokens.shape[2]),
> > ) # Line 496
> > ```
> > where `input_tokens` is the sequence of tokens to be masked and `ids_keep` is a computed mask ($M\_\text{adaptive}$ or $M\_\text{random}$ in Line 16 of Algorithm 2 in Appendix). We have used the `torch.gather` function to implement the mask operation on the sequence of tokens. For the computation details of mask, we put the code in the `enerate_random_masks_with_ref` function of `EMAC` in `emac/emac.py` (Line 179) for further reference.
> >
> >
> > **2. Will the performance be affected if different random seeds are used?**
> >
> > Thanks for your invaluable question.
> >
> > 1. In the training phase, the learnable mask tokens are initialized with the pretrained weights from MultiMAE [3]. Therefore, different random seeds will not affect the initialization of the mask tokens.
> > 2. During inference, the learnable mask token will be fixed, and the value of the mask token will not be affected by the random seeds during the inference phase. Therefore, the inference performance will not be affected by different random seeds.
> > 3. In addition, to explore the impact of different random seeds on the overall performance of our model, we have also conducted more experiments on the FDST dataset, which is a relatively small dataset for quick validations. The average MAE and RMSE scores are $1.30\_{\pm0.022}$ and $1.68\_{\pm0.015}$. The results show that different random seeds do not significantly affect the performance of our model.
> >
> >
> > |Metrics|1|2|3|Average|
> > |:-----------:| :-----------:|:-----------:|:-----------:|:-----------:|
> > |MAE| $1.29$| $1.28$ |$1.33$ |$1.30\_{\pm0.022}$|
> > |RMSE| $1.69$| $1.66$ |$1.69$|$1.68\_{\pm0.015}$|

---

> > ### Author Response · Authors · 2024-11-25
> >
> > # References
> >
> > [1] Zhang, Yingying, et al. "Single-image crowd counting via multi-column convolutional neural network." Proceedings of the IEEE conference on computer vision and pattern recognition. 2016.
> >
> > [2] Li, Yuhong, Xiaofan Zhang, and Deming Chen. "Csrnet: Dilated convolutional neural networks for understanding the highly congested scenes." Proceedings of the IEEE conference on computer vision and pattern recognition. 2018.
> >
> > [3] Bachmann, Roman, et al. "Multimae: Multi-modal multi-task masked autoencoders." European Conference on Computer Vision. Cham: Springer Nature Switzerland, 2022.

---

> > ### Author Response · Authors · 2024-11-25
> >
> > # Summary
> >
> > **We sincerely thank you once again for your insightful and detailed comments, which have been instrumental in enhancing the quality and clarity of our manuscript. In response to your feedback, we have made thorough clarifications and revisions, which we hope address your concerns comprehensively. We deeply appreciate the time and effort you have invested in providing such invaluable suggestions. If there are any additional questions or suggestions, please do not hesitate to share them with us. We remain committed to further refining the manuscript and eagerly look forward to receiving your further feedback.**

---

> > > ### Comment · Reviewer_P9ZV · 2024-11-26
> > >
> > > What is the $\mathbb{N}$ in Eq.(2)? what is the relationship among  $\mathbb{N}$ , $\mathcal{N}_I^{ret}$  and $\mathcal{P}$?

---

> ### Author Response · Authors · 2024-11-26
>
> **What is the $\mathbb{N}$ in Eq.(2)? What is the relationship among $\mathbb{N}$,
> $\mathcal{N}_I^\text{ret}$ and $\mathcal{P}$?**
>
> Thanks for your quick feedback and the invaluable questions.
>
> We denote $\mathbb{N}$ as a random variable that follows a Uniform distribution between $0$ and $1$, which is produced by a random number generator (`random.random()`), as shown in the second row of the following code. We have also updated the main paper in Line 282-283 to provide detailed descriptions.
> ```python
> if dec_order == None:   # Line 203 in emac/emac.py
>     dec_order = random.random() <= 0.8  # 0.8 is a pre-set value of 1-P
> noise_for_rgb = torch.nn.functional.unfold(
>     rearrange(ref_mask, 'b c h w->b c h w'), kernel_size=16, stride=16
> ).sum(dim=1)    # \phi_{sum}(D^i_{patch})
>
> ids_arange_shuffle_rgb = torch.argsort(
>     noise_for_rgb,
>     dim=1,
>     descending=dec_order,
> )   # argsort(\phi_{sum}(D^i_{patch}))
> ```
>
> In the code implementation (Line 203-213 in `emac/emac.py` of [code](https://anonymous.4open.science/r/E-MAC-4631/emac/emac.py)), we take $1-\mathcal{P}$ as a threshold to perform the descending order sorting, where $\mathcal{P}$ is the background retention probability. Specifically, when the value of $\mathbb{N}$ is less than $1-\mathcal{P}$, the *sorting* operation is performed in the descending order; otherwise, it is performed in the ascending order.
>
> Please kindly note that $\mathcal{N}_I^\text{ret}$ is the number of retained image tokens, and we retain the first $\mathcal{N}_I^\text{ret}$ image tokens after the *sorting*, as presented in Line 897-900 of the supplementary material.

---

> > ### Comment · Reviewer_P9ZV · 2024-11-26
> >
> > According to your response, (2) selects the last $\mathcal{N}_I^{ret}$ patches with a probability $P$ or selects the first $\mathcal{N}_I^{ret}$ patches with a probability $(1 - P)$.
> >
> > If the above is correct, I finally understand your method's pipeline. However, after understanding it, I do not notice any theoretical contributions or novel parts. Maybe this is a good submission for some conferences focusing on vision applications, but it is not acceptable for ICLR without theoretical contributions. I will increase the score to 5, and then the Ac can decide whether this is good enough to be accepted based on other reviewers' comments.
> >
> > Besides, the following comments should be further addressed if you still want to improve your paper, regardless of the score I give to this study (5 is the top score I can give to this submission): video-related models require fast processing time. I recommend the authors compare efficiency (FPS, etc.) and model size to other video counting models.

---

> > > ### Author Response · Authors · 2024-11-27
> > >
> > > Thanks for your quick reply and agreeing to raise the score. Please kindly note that our core contribution does not lie in the detailed implementation, instead, **we propose a new generative learning framework for video object counting, which first explores the potential of a large pre-trained vision foundation model for video object counting and introduces new insights to the community**. This is recognized by Reviewer 3dEQ. Besides, different from most existing methods, **we take the density map as an auxillary modality and transform the traditional density map estimation paradigm to a new multi-modal learning paradigm, ensuring the balance of fore-background effectively.** This is recognized by Reviewer Dnim and rdoG. The proposed new counting dataset presents a significant challenge associated with small object counting, offering valuable contributions to the research community, as acknowledged by most reviewers.
> > >
> > > In addition, our work focuses on a new deep model as well as a new-bulit large-scale benchmark to a specific computer vision task, which is consistent with the topic "applications to computer vision, audio, language, and other modalities" of ICLR, as the primary area we selected in our submission. It is worth noting that there are also some Outstanding and Honorable Mentioned papers in ICLR that focus on vision applications, such as [1, 2, 3], and so we believe our work meets the scope of ICLR.
> > >
> > >
> > > Finally, we have added comparisons with other video counting models and reported the FLOPs, FPS, and the number of parameters in Table 8 of the supplementary material (Line 1080-1097). All the experiments were conducted on a NVIDIA RTX 3090 GPU. It is worth noting that the FLOPs of our method and the competing methods are comparable, although we first adopted the vision foundation model. Compared to EPF and PFTF, our E-MAC achieves superior performance with fewer required computations. STGN achieves higher FPS with fewer parameters, but its performance is limited. Compared to STGN, our method achieved over $58\\%$ performance improvements with only $9\\%$ more FLOPs. These results further validate our efficiency and effectiveness.
> > >
> > > |  Model |FLOPs(G) $\downarrow$ | FPS $\uparrow$| Parameters(M) $\downarrow$ | MAE (DroneBird) $\downarrow$ | RMSE (DroneBird) $\downarrow$ |
> > > |:------:|:------:|:------:|:------:|:------:|:------:|
> > > |  EPF | 815 |19 | 20 | 97.22 | 133.01 |
> > > |  PFTF |1075 | 13 | 23 | 89.76 | 101.02 |
> > > |  STGN  |742| 30 | 13 | 92.38 | 124.67 |
> > > |  Ours |811| 16 | 98 | 38.72 | 42.92 |
> > >
> > >
> > >
> > > [1] Venkataramanan, Shashanka, et al. "Is ImageNet worth 1 video? Learning strong image encoders from 1 long unlabelled video." The Twelfth International Conference on Learning Representations (ICLR), 2024.
> > >
> > > [2] Darcet, Timothée, et al. "Vision Transformers Need Registers." The Twelfth International Conference on Learning Representations (ICLR), 2024.
> > >
> > > [3] Kim, Donggyun, et al. "Universal Few-shot Learning of Dense Prediction Tasks with Visual Token Matching." The Eleventh International Conference on Learning Representations (ICLR), 2023.

---

> > > > ### Comment · Reviewer_P9ZV · 2024-11-27
> > > >
> > > > 1. Accordingly, the authors consider the proposed model as a generative learning framework. I did not notice any part that is related to generative learning [a]. What is the definition of generative learning in your study? Additionally, there are no mentions of it in the manuscript.
> > > >
> > > > 2. Regarding density maps as a modality is not a novel insight, since previous work like [b] has a more theoretical view on it. Besides, during inference, the density map modality is not incorporated in the DEMO part, but in [b], the density map is used during the inference stage as well. Although [b] addresses prompt counting instead of video counting, the multi-modal approach (if the density map is considered as a modality) in it is more convincing. If you consider $\hat{D}_{t-1}$ in TCF as the described density map modality, it may not be novel, as this is a common operation in video-relevant tasks.
> > > >
> > > > - [a] "An Introduction to Deep Generative Modeling," Lars Ruthotto, et al., 2021.
> > > > - [b] "A Fixed-Point Approach to Unified Prompt-Based Counting," Wei Lin, et al., 2024.
> > > >
> > > > ---
> > > >
> > > > Thanks for the authors' detailed response and for presenting a nice work in video counting. However, I am not convinced by the theoretical contributions provided by the authors.

---

> > > > > ### Author Response · Authors · 2024-11-27
> > > > >
> > > > > Thanks for your feedback!
> > > > >
> > > > > 1. Our model introduces the pre-trained vision foundation model MultiMAE [1] to video object counting. The pre-trained model follows the masked modeling framework [2] and computes a pixel wise L1 or L2 loss on the reconstructed tokens, which aims to generate the masked regions, as discussed in "Probabilistic or generative modeling" of Sec. 5 in MultiMAE [1]. Please kindly note that the masked modeling is a significant subgroup of generative modeling [3], which is recognized in some recent works [4][5]. It is worth noting that the masked modeling related works are mainly proposed in and after 2022, so they did not appear in [7] (released in 2021). Based on this paradigm, we propose a density-embedded efficient masked modeling framework for video object counting, which effectively balances the fore-background information. Considering the masked modeling in our E-MAC, we believe it can be considered as a generative model, which generates the masked tokens to density map modality. To avoid confusing, the "generative learning" might be changed to "generative modeling".
> > > > >
> > > > >
> > > > > 2. We provide more explanations for our framework, the density map is used to construct the correlation between image and density map during training. The density map used in [8] is a predicted result, while we use the ground-truth density map in our framework during training. Note that, the ground-truth density map is not available during inference. As we introduced the masked modeling framework to video object counting, the density map modality can be fully masked, thus the counting model takes the video frame as input and generates the corresponding density map.
> > > > >
> > > > > While our current version might not fully meet the reviewer's standard of theoretical contributions, we appreciate the reviewer's insightful and detailed comments, which have greatly improved our work and inspired us to research more. **We also thank the reviewer for recognizing our paper as a nice work in video counting and being willing to raise the score**.
> > > > >
> > > > > ---
> > > > >
> > > > > [1] Bachmann, Roman, et al. "Multimae: Multi-modal multi-task masked autoencoders." European Conference on Computer Vision. Cham: Springer Nature Switzerland, 2022.
> > > > >
> > > > > [2] He, Kaiming, et al. "Masked autoencoders are scalable vision learners." Proceedings of the IEEE/CVF conference on computer vision and pattern recognition. 2022.
> > > > >
> > > > > [3] Kukushkin, Maksim, Martin Bogdan, and Thomas Schmid. "BiMAE-A Bimodal Masked Autoencoder Architecture for Single-Label Hyperspectral Image Classification." Proceedings of the IEEE/CVF Conference on Computer Vision and Pattern Recognition. 2024.
> > > > >
> > > > > [4] Zhang, Chaoning, et al. "A Survey on Masked Autoencoder for Visual Self-supervised Learning." IJCAI. 2023.
> > > > >
> > > > > [5] Li, Siyuan, et al. "Masked modeling for self-supervised representation learning on vision and beyond." arXiv preprint arXiv:2401.00897 (2023).
> > > > >
> > > > > [6] Bao, Hangbo, et al. "BEiT: BERT Pre-Training of Image Transformers." International Conference on Learning Representations (ICLR) 2022.
> > > > >
> > > > > [7] Lars Ruthotto, et al. "An Introduction to Deep Generative Modeling." 2021.
> > > > >
> > > > > [8] Wei Lin, et al. "A Fixed-Point Approach to Unified Prompt-Based Counting." 2024.

---

### Official Review · Reviewer_3dEQ · 2024-11-05

**Soundness:** 2
**Presentation:** 3
**Contribution:** 2
**Rating:** 6
**Confidence:** 5

**Summary:**

This paper proposes a large video bird counting dataset DroneBird, in natural scenarios for migratory bird protection. In addition, an efficient masked autoencoder for video crowd counting framework is proposed. Specifically, to integrate the foundational model and take the density map as an auxiliary modality to perform self-representation learning, a density-embedded efficient masked autoencoder counting framework is proposed. Moreover, the authors propose an efficient spatial adaptive masking method to overcome the dynamic density distribution and make the model focus on the foreground regions. Extensive experiments on three crowd datasets and our DroneBird validate the effectiveness.

**Strengths:**

Strengths:
- A large bird video counting dataset is proposed, which is helpful for bird activities analysis;
- Applying masked authencoding learning for video crowd counting, which is interesting;
- The proposed approach achieves SOTA video counting performance on several benchmarks, especially for the proposed bird dataset;

**Weaknesses:**

Weaknesses:
- Ambiguous performance gains: The proposed approach integrates a masked autoencoder for representation learning and formulates the previous density regression task as "density reconstruction" within an MAE-like framework for crowd counting. Additionally, the authors use the pre-trained ViT-B from MultiMAE as the encoder for discriminative feature extraction. However, given that pre-trained MAE models typically lead to significant performance improvements during downstream task fine-tuning, it remains unclear whether the observed performance gains stem from the designed new masked autoencoder-based framework or the usage of pre-trained MultiMAE. One suggestion: compared w/ a variant w/o using the pre-trained ViT;
- For the proposed Temporal Collaborative Fusion, it is unclear why we need to calculate $\hat{D}_{t}^{res}$ again after obtaining $\hat{D}_{t}$ and $\hat{D}_{t-1}$ from the decoder. If $\hat{D}_{t}$ and $\hat{D}_{t-1}$ are not accurate enough, then $\hat{D}_{t}^{res}$ calculated using them will also be inaccurate. Conversely, if $\hat{D}_{t}$ and $\hat{D}_{t-1}$ are accurate, there seems to be no need for the Temporal Collaborative Fusion module.
- A comparison of model complexity with the other approaches in Table 1 should be provided to ensure a fair evaluation, as the current work may use a relatively larger model.

**Questions:**

- One open question: As we all know that MAE pre-training can improve many downstream task fine-tuning. For video crowd counting, does the current approaches also benefit from the MAE pre-trained models (e.g., MAE ViT-base)? i.e., using them as the initialization weights for video counting fine-tuning.

As mentioned in the Weaknesses, my main concerns lie in the ambiguous performance gains and methodology design. Since the proposed approach uses the pre-trained MultiMAE model and jointly follows a masked autoencoder framework for fine-tuning, it is not clear whether the performance improvements stem from the pre-trained model or downstream masked fine-tuning with the designed framework from the authors.  I would like to see the author rebuttal in terms of this part.

---

> ### Author Response · Authors · 2024-11-22
> **Response to Reviewer 3dEQ (1/4)**
>
> We thank the reviewer for recognizing our **helpful bird counting dataset**, **interesting framework** and **SOTA performance**. We appreciate your support and constructive suggestions and address your concerns as follows.
>
> **Weakness 1: Ambiguous performance gains: The proposed approach integrates a masked autoencoder for representation learning and formulates the previous density regression task as "density reconstruction" within an MAE-like framework for crowd counting. Additionally, the authors use the pre-trained ViT-B from MultiMAE as the encoder for discriminative feature extraction. However, given that pre-trained MAE models typically lead to significant performance improvements during downstream task fine-tuning, it remains unclear whether the observed performance gains stem from the designed new masked autoencoder-based framework or the usage of pre-trained MultiMAE. One suggestion: compared w/ a variant w/o using the pre-trained ViT.**
>
> Thanks for the valuable comment. We provide more explanations for the effect of the proposed new masked autoencoder-based framework in two aspects:
>
> 1. **It is difficult to train a well-performing ViT from scratch using the existing counting dataset.** Please kindly note that using the pre-trained ViT alone is hard to achieve considerable counting performance. We have compared the performance of vanilla ViT [1] with the existing method (PET [2], Table 1 in Sec.5.2 of the main paper) in the Table below. As a comparison, vanilla ViT (Exp. 2 in Table below) is significantly worse than PET [2], which demonstrates the contribution of solely using a pre-trained model is limited. The experiments with pre-trained MultiMAE [3] (Exp. 3 in Table below / Exp. I in Table 2 of the main paper) also show similar results. Instead, the proposed E-MAC performed based on the pre-trained model achieves 1.29 and 1.69 of MAE and RMSE, which is an unprecedented improvement of 47% over MultiMAE. This fully demonstrates the effectiveness of our method.
>  2. Besides, we have added experiments on vanilla ViT (Exp. 2 in the Table below) and E-MAC w/ vanilla ViT pre-trained weights (Exp. 4 in the Table below). Compared with the ablated experiment Exp. I and Exp. V (Table 2 in Sec.5.3 of the main paper), although using the pre-trained MultiMAE weights further enhances the performance, **the performance improvement of our method mainly comes from our uniquely designed E-MAC**. These experiments validate our effectiveness.
>
>     It is worth noting that we first introduce the pre-trained vision model to the video counting object task, and propose a novel E-MAC framework, which treats the density map as an auxiliary modality to guide the mask modeling, achieving superior performance and offering a new perspective to the community.
>
>     |No.|  Model | MAE $\downarrow$  | RMSE $\downarrow$ |
>     |:------:|:------:|:------:|:------:|
>     |1|  PET [2] | 1.73 | 2.27 |
>     |2|  vanilla ViT [1]  |  2.63  | 3.31 |
>     |3|  MultiMAE (Exp. I in Sec.5.3) | 2.45 | 3.22 |
>     |4|  E-MAC w/ vanilla ViT | 1.49 | 1.93 |
>     |5|  E-MAC w/ MultiMAE (Exp. V in Sec.5.3)  |  1.29  | 1.69 |

---

> ### Author Response · Authors · 2024-11-22
> **Response to Reviewer 3dEQ (2/4)**
>
> **Weakness 2: For the proposed Temporal Collaborative Fusion, it is unclear why we need to calculate $\hat{D}^{res}\_t$ again after obtaining $\hat{D}\_{t}$ and $\hat{D}\_{t-1}$ from the decoder. If $\hat{D}\_{t}$ and $\hat{D}\_{t-1}$ are not accurate enough, then $\hat{D}\_{t}^{res}$ calculated using them will also be inaccurate. Conversely, if $\hat{D}\_{t}$ and $\hat{D}\_{t-1}$ are accurate, there seems to be no need for the Temporal Collaborative Fusion Module.**
>
> Thanks for the valuable comment. $\hat{D}\_{t}^{res}$ is predicted by a cross-attention layer, in which $\hat{D}\_{t}$ is taken as query and $\hat{D}\_{t-1}$ is taken as key and value. Please kindly note that **$\hat{D}\_{t}^{res}$ is designed to provide temporal information for $\hat{D}\_{t}$**.
>
> More specifically, due to the potential inaccurate prediction for some extreme scenarios, we proposed the Temporal Collaborative Fusion model to improve the performance. Actually, the **Robustness and accuracy can be effectively improved through multi-frame ensemble**, which is a generally adopted approach [4,5,6].  Our TCF aims to provide corrections to the original prediction results to improve accuracy. Different frames may contain more comprehensive information, which could be integrated to produce more accurate results. We provide a visual analysis of the results of the TCF in Sec. A.5 of the Appendix, from which it is clear that the TCF removes some of the background interference. We have also added an experiment to compare the performance of E-MAC with and without TCF, as shown in the table below, the addition of TCF improves performance by 15%. Therefore the design of TCF is reasonable and effective.
>
> |  Model | MAE $\downarrow$ | RMSE $\downarrow$ |
> |:------:|:------:|:------:|
> |  E-MAC w/o TCF  |  1.51  | 2.03 |
> |  E-MAC w/ TCF (Exp.V in Sec.5.3)  |  1.29  | 1.69 |

---

> ### Author Response · Authors · 2024-11-22
> **Response to Reviewer 3dEQ (3/4)**
>
> **Weakness 3: A comparison of model complexity with the other approaches in Table 1 should be provided to ensure a fair evaluation, as the current work may use a relatively larger model.**
>
> Thanks for the valuable comment. We have compared the number of parameters and FLOPs with some previous methods and reported the results in the Table below. **It is worth noting that the FLOPs of our method and the competing methods are comparable**, although we first adopted the vision foundation model. Compared to EPF [7] and PFTF [8], E-MAC achieves superior performance with less required computations. Compared to STGN [9], our method achieved a 58% performance improvement with only 9% more FLOPs.
>
>
> |  Model | FLOPs(G) $\downarrow$ | Parameters(M) $\downarrow$ | MAE (DroneBird) $\downarrow$ | RMSE (DroneBird) $\downarrow$ |
> |:------:|:------:|:------:|:------:|:------:|
> |  EPF | 815 | 20 | 97.22 | 133.01 |
> |  PFTF | 1075 | 23 | 89.76 | 101.02 |
> |  STGN | 742 | 13 | 92.38 | 124.67 |
> |  E-MAC| 811 | 98 | 38.72 | 42.92 |

---

> ### Author Response · Authors · 2024-11-22
> **Response to Reviewer 3dEQ (4/4)**
>
> **Question 1: As we all know that MAE pre-training can improve many downstream task fine-tuning. For video crowd counting, does the current approaches also benefit from the MAE pre-trained models (e.g., MAE ViT-base)? i.e., using them as the initialization weights for video counting fine-tuning.**
>
> Thanks for the valuable question. We have provided a comparison of the performance of our model and the vanilla ViT model when fine-tuning w/ and w/o the MultiMAE pretrained weights.
> 1. Fine-tuning with pretrained weights indeed improves performance, but the improvement is limited (Exp. 1 *vs.* Exp. 2 and Exp. 3 *vs.* Exp. 4 in the Table below).
> 2. In contrast, our unique design tailored to the data and video counting tasks significantly enhances performance (Exp. 1 *vs.* Exp. 3 and Exp. 2 *vs.* Exp. 4 in the Table below).
> 3. We attempted to replace the backbones of some existing methods [9,10] with pre-trained ViT, but observed varying degrees of decline in performance metrics (Exp. 5 and 6 in the Table below). This might be due to the large number of parameters in ViT, and the existing methods were not specifically designed to accommodate this.
>
> Our approach, however, efficiently learns through density map representation learning with DEMO and foreground-background balance using SAM. Please kindly note that **our method is the first work to explore the potential of a pre-trained vision foundation model for video object counting**, which provides a new perspective to the community.
>
> |Exp.|  Model | MAE $\downarrow$  | RMSE $\downarrow$ |
> |:------:|:------:|:------:|:------:|
> |1|  vanilla ViT [1]  |  2.63  | 3.31 |
> |2|  MultiMAE (Exp. I in Sec.5.3) | 2.45 | 3.22 |
> |3|  E-MAC w/ vanilla ViT | 1.49 | 1.93 |
> |4|  E-MAC w/ MultiMAE (Exp. V in Sec.5.3)  |  1.29  | 1.69 |
> |5| STGN w/ MultiMAE | 4.04| 4.19 |
> |6| Gramformer w/ MultiMAE | 5.15 | 6.32 |

---

> > ### Comment · Reviewer_3dEQ · 2024-11-25
> > **Response to Authors**
> >
> > I appreciate the authors provide additional experimental results and detailed explanation in the rebuttal, which address most of my concerns, especially for ambiguous performance gains and the module design. The work explores the potential of a large pre-trained vision foundation model for video object counting, which introduces new insights to the community in terms of generative counting model design. Thus I improve my rating.

---

> > > ### Author Response · Authors · 2024-11-26
> > > **Thank you!**
> > >
> > > Thanks a lot for your positive reply and recognizing our new insights to the community in terms of generative counting model design. We are also delighted to have addressed most of your concerns and deeply appreciate your insightful questions and comments, which have significantly enhanced the quality of our paper. Should you have any further questions or suggestions, please do not hesitate to post them. We remain committed to actively addressing your concerns and continuously refining this work.

---

> ### Author Response · Authors · 2024-11-22
>
> # References
> [1] Dosovitskiy, Alexey. "An image is worth 16x16 words: Transformers for image recognition at scale." arXiv preprint arXiv:2010.11929 (2020).
>
> [2] Liu, Chengxin, et al. "Point-query quadtree for crowd counting, localization, and more." Proceedings of the IEEE/CVF International Conference on Computer Vision. 2023.
>
> [3] Bachmann, Roman, et al. "Multimae: Multi-modal multi-task masked autoencoders." European Conference on Computer Vision. Cham: Springer Nature Switzerland, 2022.
>
> [4] Pengyu Zhang, Jie Zhao, Dong Wang, Huchuan Lu, Xiang Ruan; Proceedings of the IEEE/CVF Conference on Computer Vision and Pattern Recognition (CVPR), 2022, pp. 8886-8895
>
> [5] Zhang, Lu, et al. "Weakly aligned cross-modal learning for multispectral pedestrian detection." Proceedings of the IEEE/CVF international conference on computer vision. 2019.
>
> [6] Kappeler, Armin, et al. "Video super-resolution with convolutional neural networks." IEEE transactions on computational imaging 2.2 (2016): 109-122.
>
> [7] Liu, Weizhe, Mathieu Salzmann, and Pascal Fua. "Estimating people flows to better count them in crowded scenes." Computer Vision–ECCV 2020: 16th European Conference, Glasgow, UK, August 23–28, 2020, Proceedings, Part XV 16. Springer International Publishing, 2020.
>
> [8] Avvenuti, Marco, et al. "A spatio-temporal attentive network for video-based crowd counting." 2022 IEEE Symposium on Computers and Communications (ISCC). IEEE, 2022.
>
> [9] Wu, Zhe, et al. "Spatial-temporal graph network for video crowd counting." IEEE Transactions on Circuits and Systems for Video Technology 33.1 (2022): 228-241.
>
> [10] Lin, Hui, et al. "Gramformer: Learning Crowd Counting via Graph-Modulated Transformer." Proceedings of the AAAI Conference on Artificial Intelligence. Vol. 38. No. 4. 2024.

---

### Author Response · Authors · 2024-11-22
**Rebuttal by Authors**

Dear PCs, SACs, ACs, and Reviewers,

We would like to than you for your valuable feedback and insightful reviews, which have greatly contributed to imporving the paper. In this paper, we proposed an **interesting** (Reviewer 3dEQ) framework for video object counting, which **tackles the imbalance of fore-background effectively** (Reviewer Dnim) and **handles dynamic density distribution** (Reviewer Dnim), experiments on multiple dataset validata E-MAC's **efficiency**  (Reviewer rdoG) and **feasible** (Reviewer rdoG). We also proposed a **new** (Reviewer P9ZV) **novel** (Reviewer rdoG) and **helpful bird counting dataset** (Reviewer 3dEQ), which **significantly broadens the application domain** (Reviewer Dnim).

We hope that our responses will satisfactorily address your questions and concerns. We address the reviewers' concerns with the following updates and improvements:

1. **Extended Analysis and Experiments**: More ablation studies including the impact of pretrained weight, effect of $\texttt{TCF}$ and visual analyses of the key components ($\texttt{TCF}, \texttt{DEMO}, \texttt{SAM}$).
2. **Detailed Descriptions**: More detailed descriptions of the key components ($\texttt{TCF}, \texttt{DEMO}, \texttt{SAM}$) and more detailed framework diagram.
3. **More dataset analysis**: Detailed analysis of the DroneBird dataset, including the comparison with existing datasets and the challenges of the new dataset.
4. **Point-by-Point Responses**: Comprehensive responses to specific reviewer comments.

We sincerely appreciate the time and effort you have dedicated to reviewing our submission, along with your invaluable suggestions. If you have any other concers, please feell free to contact us. We look forward to discussing with you. We have released an anonymous [URL](https://anonymous.4open.science/r/E-MAC-4631/README.md) with our code of model for better understanding, and we have also prepared an updated manuscript and supplementary material that addresses the reviewers' concerns.

Sincerely,

The Authors

---

### Author Response · Authors · 2024-12-02
**A gentle reminder for the close of the author-reviewer discussion & summary of the rebuttal**

Dear PCs, SACs, ACs, and Reviewers,

As the author-reviewer discussion period is closing at the end on December 2nd, we would like to call for any further discussion or comments on our submission.

In our rebuttal, we addressed the following raised concerns/misunderstandings:

- We have provided more experimental results and visualizations for validating each component.
- We have provided more explanations for the pipeline of E-MAC and more details for some figures.
- We have added experiments on the efficiency of our method.
- We have added comparisons with some competing methods on different pre-trained models.
- We have clarified the new challenges of our DroneBird dataset for video counting.
- We have carefully checked and revised all the representation issues throughout the paper.

We believe that these clarifications and additional details strengthen our paper and address the reviewers' concerns. We would like to thank the reviewer for recognizing our work **tackling the imbalance of fore-background effectively** and that our dataset **significantly broadens the application domain** (Reviewer Dnim). We are also delighted that the reviewers believe our responses have **addressed their concerns** (Reviewer 3dEQ, rdoG), and consider our submission to be **a nice work in video counting** (Reviewer P9ZV).

We understand the constraints of time and workload that reviewers and AC face, and we appreciate the effort already put into evaluating our work. If there are any additional insights, questions, or clarifications on our responses/submission that you would like to discuss with us, we would be very grateful to hear them, your feedback is invaluable for the improvement of our research.


Best regards,

Authors of submission 678

---

### Meta-Review · Area_Chair_xwMA · 2024-12-19

**Metareview:**

The paper initially received mixed reviews 5563. The major concerns were:

1. missing ablation study showing the effect of the Multi-MAE backbone [3dEQ]
2. missing comparison of model complexity [3dEQ]
3. motivation for TCF, and needs better description? [3dEQ, P9ZV]
4. comparison using MAE for other video counting methods [3dEQ]
5. how to do inference without GT input? [xwMA]
6. are experiments using the same backbone in Table 1? [Dnim]
7. method components lack cohesive integration [rdoG]
8. what is the value of the new dataset? [rdoG]
9. video-based methods underperform image ones on DroneBird, why? [rdoG]

The authors wrote a response, and had significant discussions with the reviewers. After the discussion, ratings were increased to 6865. The remaining negative reviewer thought that there was little theoretical contribution and incremental novelty, but also conceded that the proposed method could be used in other areas or vision applications, such as tiny object detection and segmentation, and thus could generally be interesting to the community.

Overall, the reviewers noted that the use of MAE architecture for video crowd counting was interesting, and achieved SOTA results. The AC agrees and recommend accept. The authors should revise the paper according to the reviews, response, and helpful discussions.

**Additional Comments On Reviewer Discussion:**

see above

---

### Decision · Program_Chairs · 2025-01-22

Accept (Poster)